# SEMREP: Generative Code Representation Learning with Code Transformations

Weichen Li [1]    Jiamin Song [1]    Bogdan Alexandru Stoica [2]    Arav Dhoot [3]    Gabriel Ryan [4]    Shengyu Fu [4]    Kexin Pei [1]

## Abstract

Code transformation is a foundational capability in the software development process, where its effectiveness relies on constructing a high-quality code representation to characterize the input code semantics and guide the transformation. Existing approaches treat code transformation as an end-to-end learning task, leaving the construction of the representation needed for semantic reasoning implicit in model weights or relying on rigid compiler-level abstractions. We present SEMREP, a framework that improves code transformation through *generative code representation learning*. Our key insight is to employ the semantics-preserving transformations as the intermediate representation, which serves as both a generative mid-training task and the guidance for subsequent instruction-specific code transformations. Across general code editing and optimization tasks (e.g., GPU kernel optimization), SEMREP outperforms the extensively finetuned baselines with strictly the same training budget by 6.9% in correctness, 1.1× in performance, 13.9% in generalization, and 6.7% in robustness. With the improved exploration of diverse code transformations, SEMREP is particularly amenable to evolutionary search. Combined with an evolutionary coding agent, SEMREP finds optimizations that 685B larger-weight baselines fail to discover while achieving the same performance with 25% less inference compute.

## 1. Introduction

Code transformation is a foundational step in the software engineering process, enabling a wide variety of software development and maintenance workflows. To automate

[1]The University of Chicago [2]University of Illinois Urbana-Champaign [3]Columbia University [4]Microsoft. Correspondence to: Weichen Li <weichenli@uchicago.edu>, Kexin Pei <kpei@uchicago.edu>.

*Proceedings of the 43rd International Conference on Machine Learning*, Seoul, South Korea. PMLR 306, 2026. Copyright 2026 by the author(s).

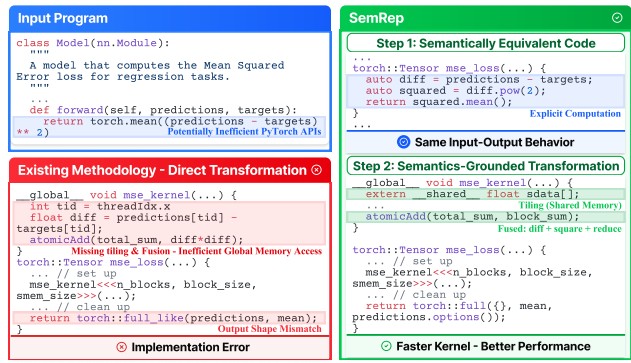

*Figure 1.* A concrete example showing how SEMREP optimizes CUDA kernels in Kernelbench (Ouyang et al., 2025). SEMREP first generates a semantically equivalent code that explicitly exposes the computation of MSE loss, and then transforms it into a more efficient implementation that optimizes memory access by replacing separate passes over large arrays with a single pass, where each thread computes its global index, reads inputs into fast registers for local computation, and fuses multiple operators into a single kernel execution. The existing approach attempts to directly optimize MSE but fails on several core operations.

code transformation, Large Language Models (LLMs) have been increasingly adopted to assist developers in concrete code editing tasks, such as efficiency optimization (Shetty et al., 2026; Ma et al., 2025; He et al., 2025), refactoring and migration (Ziftci et al., 2025; Eniser et al., 2024), bug fixing (Jimenez et al., 2024; Yu et al., 2025), and feature integration (Chi et al., 2025; Cassano et al., 2024; Nam et al., 2025). Emerging evidence also suggests that LLM-driven code transformation can discover high-performance GPU kernels (Ouyang et al., 2025; Wei et al., 2025), co-design system architectures (Cheng et al., 2025a;b), and even serve as a substrate for algorithmic and scientific discovery (Press et al., 2026; Novikov et al., 2025; Lange et al., 2025).

A key capability for effective code transformation is to disentangle *what must be preserved* from *what can be changed*. This often requires the model to understand the intended behavior of the original code, i.e., its semantics, and to reason about the effects introduced by the transformation. Figure 1 illustrates why this matters: without recognizing that the input code performs a matrix reduction, state-of-the-art approaches (Baronio et al., 2025) generate incorrect edits while missing the optimization opportunity to fuse

operations and optimize memory access patterns.

To characterize the intended behavior of the original code and support desired transformations, automated tools often explore intermediate code *representations* to expose the code properties, such as those used in compilers like abstract syntax trees, or control/data flow graphs, where a fixed set of compiler transformations on top of these representations is well defined and can be efficiently implemented.

Unfortunately, when extending the support of compiler transformations to broader natural language-instructed code editing, existing works often degenerate to be entirely data-driven, treating the transformation as an end-to-end learning task while relying on the model to *implicitly learn representations necessary to code transformation as latent weight parameters* (Shypula et al., 2023; Tan et al., 2024; Wei et al., 2026). While some works explicitly incorporate more structured representations in the model (Allamanis et al., 2018; Pei et al., 2023), constructing such representations on the fly can be expensive. More importantly, it is unclear whether these rigid representations remain *informative* for code LLMs (which are predominantly pre-trained on source code rather than compiler-level abstractions), and *optimal*, as a single structural representation suitable for transforming an input code may not necessarily generalize to another.

**Our approach.** We present SEMREP, a new framework that improves code transformation capabilities by training LLMs to explicitly learn code representations as semantics-preserving transformations. Specifically, we define a mid-training task, *generative code representation learning*, by training the model to generate semantically equivalent programs, i.e., producing the same execution outputs as the original program for given inputs (Le et al., 2014). During inference, given the input code, the trained model is allowed to alternate between the generation of (1) semantically equivalent code snippets as part of the *exploration*, and (2) transformed code following the initial instructions, based on the code generated in (1). Figure 1 shows how SEMREP generates a single semantically equivalent code, which inspires the generation of the optimized CUDA kernel. Figure 2 shows its overall workflow.

SEMREP's task formulation encourages the model to produce explicit code representations, i.e., semantically equivalent source code, that remain directly *interpretable* by LLMs and humans, but also amenable to *unambiguous* code reasoning tools, such as execution (Ding et al., 2024; Copet et al., 2025) and static analysis (van Tonder & Le Goues, 2020; Chen et al., 2018; Namjoshi & Pavlinovic, 2018). The former enables the generation of *diverse and human-interpretable structured reasoning*, i.e., chain of code (Li et al., 2023), to enable the generation of (potentially more optimal) subsequent instruction-specific code transformation, while the latter ensures such a structured reasoning is

*verifiably correct* (modulo test input (Le et al., 2014)).

Such a design offers several benefits when guiding LLMs' training and inference for code transformation. First, it teaches the model to discover representations that express code semantics *explicitly*, as opposed to latent representation in model weights. Second, it exposes opportunities for *diverse* and *exploratory* code transformations, which are particularly amenable for test-time scaling like evolutionary search, where exploration is the key to finding an optimal solution (Novikov et al., 2025). Third, the verifiable rewards are particularly well-suited to reinforcement learning, sharing a similar spirit to reinforcement pre-training (Dong et al., 2025; Hatamizadeh et al., 2025) that is *easy to scale*. Lastly, the generative formulation of semantically equivalent code often aligns with most of the practical code editing setups, where it is required to preserve the original code functionality while introducing new properties, such as performance, readability, and security (Li et al., 2025).

We implement our representation mid-training using reinforcement learning (RL) with verifiable rewards (Guo et al., 2025; Shao et al., 2024) based on test execution, and then finetune (via RL) the model for various code transformation tasks like CUDA kernel optimizations (Ouyang et al., 2025) and general-purpose code editing (Chi et al., 2025). To show the unique advantage of the generative representation learning, we construct a dedicated baseline that adopts the same model as SEMREP but only finetuned for the code transformation (not mid-trained) with the *exactly same training budget*, i.e., the total iterations of mid-training and finetuning in SEMREP matches this baseline's finetuning iterations. This ensures that we do not introduce extra compute when training for representation learning.

**Results.** Our evaluation shows that SEMREP enables smaller models, e.g., QwQ-32B (Qwen, 2025), to match or outperform models with $12\times$ larger weights, and commercial LLMs, across all tasks and inference paradigms. SEMREP outperforms the extensively finetuned baselines with strictly the same training budget by 6.9% in correctness and $1.1\times$ in performance. With explicit representation learning, SEMREP significantly improves the baseline in generalization, e.g., by up to 13.9% in cross-hardware kernel optimization, and 6.7% improved robustness against semantics-preserving code transformations. When integrating SEMREP with advanced evolutionary coding agent (Sharma, 2025), it is able to find optimization that strong baselines (e.g., DeepSeek-V3-Reasoner) fail to find, while achieving the same performance with 25% less compute.

## 2. Methodology

**Problem statement.** We formulate the code editing task as the transformation of a source program $C_{src}$ into a target

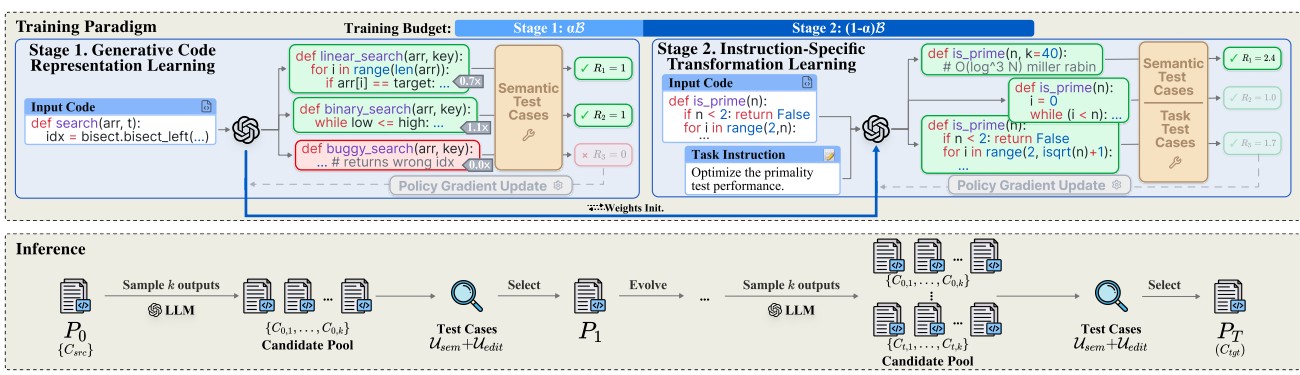

*Figure 2.* SEMREP framework. The upper shows how SEMREP trains a model to instill semantics understanding through the generative code representation learning. SEMREP explicitly encourages diverse semantically equivalent implementations, e.g., both faster and slower code variants, even when editing objectives are not fully satisfied. The total training budget $\mathcal{B}$ is fixed to ensure the fair comparison to those finetuning-only baselines. $\alpha$ modulates the training budget allocated to Stage 1 and Stage 2. The lower figure shows how SEMREP performs iterative inference.

program $C_{tgt}$ conditioned on a natural language instruction $I$. This transformation must meet two distinct requirements: (1) semantic preservation of existing logic, and (2) alignment with the new instruction.

We define the semantics-preserving transformation using test-based equivalence (Le et al., 2014). Given a set of tests $\mathcal{U}$, two programs $C_1$ and $C_2$ are considered equivalent modulo the tests $\mathcal{U}$, denoted as $C_1(\mathcal{U}) = C_2(\mathcal{U})$, if they produce identical outputs for all tests in $\mathcal{U}$.

We use two test sets, $\mathcal{U}_{sem}$ and $\mathcal{U}_{edit}$, to measure semantic preservation and instruction adherence, respectively. The goal is to learn a conditional distribution $P(C_{tgt}|C_{src}, I)$, subject to $C_{src}(\mathcal{U}_{sem}) = C_{tgt}(\mathcal{U}_{sem})$, and $C_{tgt}$ satisfies the requirements specified in $I$ that is verifiable by $\mathcal{U}_{edit}$.

The challenge in code editing is to disentangle what needs to be preserved (e.g., the original functionality of $C_{src}$) with respect to $\mathcal{U}_{sem}$, and what needs to be edited (the new requirement specified in $I$) with respect to $\mathcal{U}_{edit}$.

To address this challenge, SEMREP decomposes the editing process into the transition: $C_{src} \rightarrow \cdots \rightarrow C_{rep} \rightarrow \cdots \rightarrow C_{tgt}$, where $C_{rep}$ serves as an explicit and verifiable intermediate semantic representation that satisfies $C_{rep}(\mathcal{U}_{sem}) = C_{src}(\mathcal{U}_{sem})$, before further guided changes are applied. These transitions are learned in two stages: (1) Generative code representation learning trains the model to generate semantically equivalent program variants. (2) Instruction-specific transformation finetuning trains the model to transform programs to align with $I$.

### 2.1. Generative Code Representation Learning

In the first stage of SEMREP's training, we optimize a base model to generate semantics-preserving variants of $C_{src}$. The goal is to produce a representation $C_{rep}$ that explores the semantic equivalent class of $C_{src}$ that stays function-

ally invariant under the tests $\mathcal{U}_{sem}$. We formulate this as a reinforcement learning task using Group Relative Policy Optimization (GRPO) (Shao et al., 2024), and define the reward $R_{sem}$ that measures semantic invariance while encouraging the model to explore different syntax or structures:

$$R_{sem}(C_{rep}) = \alpha_1 \cdot \mathbb{I}_{comp}[C_{rep}] + \beta_1 \cdot \mathbb{I}_{\mathcal{U}_{sem}}[C_{rep} = C_{src}]$$

Here $\alpha_1$ and $\beta_1$ are weights that modulate the focus between compilability (as it is a pre-requisite for the code to be executable using $\mathcal{U}_{sem}$) and semantic invariance. $\mathbb{I}_{comp}(C_{rep})$ denotes successful compilation, and $\mathbb{I}_{\mathcal{U}_{sem}}[C_{rep} = C_{src}]$ is the indicator function for equivalence based on the test set $\mathcal{U}_{sem}$. This reward design is intended to encourage exploring diverse, semantically equivalent variants of the given source code during training (e.g., different loop structures or API calls) that elicit creative transformation possibilities in the downstream editing task, e.g., code optimizations.

### 2.2. Instruction-Specific Transformation Learning

In the second stage of SEMREP's training, we use an instruction-specific editing objective, where the model must introduce the changes requested by $I$ while preserving the original program's intended behavior unaffected by the edit.

Similar to Section 2.1, we employ GRPO and optimize for an instruction-specific reward $R_{inst}$ that jointly rewards instruction-following and semantics-preserving changes in the transformed program:

$$R_{inst}(C_{tgt}) = \alpha_2 \cdot \mathbb{I}_{comp}[C_{tgt}] + \beta_2 \cdot \mathbb{I}_{\mathcal{U}_{sem}}[C_{src} = C_{tgt}] + \gamma \cdot \mathbb{I}_{\mathcal{U}_{edit}}[C_{tgt}]$$

Here $\alpha_2, \beta_2, \gamma$ control the compilability, semantic invariance, and instruction adherence. $\mathbb{I}_{comp}[C_{tgt}]$ denotes suc-

cessful compilation, $\mathbb{I}_{\mathcal{U}_{sem}}[C_{src} = C_{tgt}]$ measures equivalence, and $\mathbb{I}_{\mathcal{U}_{edit}}[C_{tgt}]$ verifies instruction adherence.

## 2.3. Iterative Inference and Test-Time Scaling

We scale the inference by an iterative evolutionary search, where the model alternates between semantic exploration, i.e., generating diverse semantically equivalent code, and instruction-specific transformation, i.e., optimization, feature integration, etc. This generates a sequence of program pools $P_0, P_1, \ldots, P_T$, where $P_0 = \{C_{src}\}$, and $\forall t \in [1, T], P_t = \{c_1, \cdots, c_b\}$ maintains a beam that contains $b$ programs, i.e., number of programs retained at the end of each iteration. During each turn $t \in [1, T]$, we sample $k$ transformed candidates for each program in $P_{t-1}$ by querying the model, resulting in $k \cdot b$ candidate programs: $\tilde{P}_t = \{\tilde{c}_1, \ldots, \tilde{c}_{k \cdot b}\}$. From $\tilde{P}_t$, we keep only top-$b$ selected programs based on their execution against $\mathcal{U}_{sem}$ and $\mathcal{U}_{edit}$:

$$P_t = \underset{c \in \tilde{P}_t}{\text{Top-}b}\big[\omega_1 \mathcal{U}_{sem}[c = C_{src}] + \omega_2 \mathcal{U}_{edit}(c)\big]$$

Here $\omega_1$ and $\omega_2$ are weights that modulate the focus between semantic exploration (finding functionally equivalent variants) and instruction-following editing (realizing editing requests). Importantly, the capability of generating semantic-preserving programs (not necessarily optimized or fully aligned with user instructions) as intermediate steps facilitates the discovery of more source programs that could potentially lead to more effective edits in downstream tasks.

The final target program $C_{tgt}$ is sampled from $k$ trajectories maintained across $T$ iterations $P_1, \ldots, P_T$, and selected based on their execution against $\mathcal{U}_{sem}$ and $\mathcal{U}_{edit}$, e.g., ranked in speedup for optimizations with functionality guarantee.

## 3. Evaluation

We evaluate SEMREP on two critical software engineering applications that can be formulated as code editing tasks (Section 3.1). We consider inference using the finetuned model as our default editing mode and compare it to the state-of-the-art baselines, which are also mostly based on finetuned models (Baronio et al., 2025). We choose models that can be full-parameter RL-finetuned with FSDP (Zhao et al., 2023) on our local hardware (2x4 Nvidia L40S GPUs), i.e., Qwen2.5-Coder-7B (Hui et al., 2024), and QwQ-32B (Qwen, 2025). We also include multiple case studies on KernelBench (Appendix D), Editbench (Section 3.6), integration with evolutionary coding agent (Section 3.7), and repo-level analysis (Appendix E).

## 3.1. Setup: Tasks, Datasets and Metrics

**Tasks and benchmarks.** We evaluate SEMREP on Edit-Bench (Chi et al., 2025) and KernelBench (Ouyang et al.,

*Table 1.* Resolve rate (pass@1, %) of SEMREP and other representative models on EditBench (Chi et al., 2025) leaderboard using Core set. SEMREP uses QwQ-32B as a base model, and we report the single run for comparability.

| Model | Model Size | Pass@1 ↑ |
|---|---|---|
| *Closed Weight Models* | | |
| Claude Sonnet 4 | - | 66.67 |
| GPT o4-mini | - | 57.41 |
| Gemini 2.5 Pro | - | 54.63 |
| *Open Weight Models* | | |
| SEMREP QwQ-32B (Ours) | 32B | **57.41** |
| Qwen3-Coder | 405B | 55.56 |
| GLM-4.6 | 355B | 55.56 |
| Finetuned QwQ-32B | 32B | 53.70 |
| Qwen2.5-72B-Instruct | 72B | 53.70 |
| QwQ-32B | 32B | 50.93 |
| SEMREP Qwen2.5-Coder-7B | 7B | 32.41 |
| gemma-3-12b-it | 12B | 23.15 |
| Finetuned Qwen2.5-Coder-7B | 7B | 23.15 |
| Qwen2.5-Coder-7B | 7B | 18.52 |

2025). EditBench is constructed from real-world repo-level edits to evaluate how well LLMs follow developer instructions (in natural language) to perform diverse software maintenance operations, e.g., bug fixes, feature modifications or enhancements, and code refactoring. KernelBench focuses on optimizing PyTorch implementations with customized low-level CUDA kernels to improve performance.

**Metrics.** For EditBench, we report performance using the primary metric established in the leaderboard: *Pass@k*: The percentage of problems where one of the model's $k$ generated edits passes all associated unit tests. This provides a rigorous assessment of functional correctness, evaluating the model's ability to implement intended changes and produce deployable code on the first attempt.

For KernelBench, we adopt the same metrics as Kevin (Baronio et al., 2025): *Correctness*: whether an LM-generated trajectory contains at least one correct kernel. *Speedup*: the absolute ratio between the execution time required by the given PyTorch code and the fastest and correct code within an LM-generated trajectory. *$fast_p$*: whether there exists a correct code that achieves speedup $> p$ within the given trajectory. For each metric, we report two aggregations across $k$ parallel trajectories: *Best@k*: The maximum value achieved across $k$ trajectories. *Avg@k*: The arithmetic mean of the metric across all $k$ trajectories.

**Training and inference.** We train SEMREP with verl (Sheng et al., 2025) and use GRPO (Shao et al., 2024) without a KL penalty following Baronio et al. (2025); Hatamizadeh et al. (2025). To ensure a fair comparison, we enforce a *strictly same* training budget across all ex-

*Table 2.* Main results for domain-specific code editing on KernelBench.

| | Correctness↑ | | Speedup↑ | | fast$_1^*$ ↑ | | fast$_{1.5}^*$ ↑ | |
|---|---|---|---|---|---|---|---|---|
| | best@16 | avg@16 | best@16 | avg@16 | best@16 | avg@16 | best@16 | avg@16 |
| QwQ-32B | 33 | 4.13 | 1.25 | 1.02 | 14 | 2.06 | 4 | 0.69 |
| GPT 4o-mini | 47$_{+14}$ | 17.13$_{+13}$ | 1.27$_{+0.02}$ | 1.12$_{+0.1}$ | 22$_{+8}$ | 7.75$_{+5.69}$ | 8$_{+4}$ | 2.13$_{+1.44}$ |
| Kevin-32B | 65$_{+32}$ | 19.63$_{+15.5}$ | 1.36$_{+0.11}$ | 1.1$_{+0.08}$ | 21$_{+7}$ | 25$_{+22.94}$ | 9$_{+5}$ | 2.69$_{+2}$ |
| **SEMREP (Ours)** | **93**$_{+60}$ | **37.63**$_{+33.5}$ | **2.87**$_{+1.62}$ | **1.21**$_{+0.19}$ | **72**$_{+58}$ | **21.06**$_{+19}$ | **12**$_{+8}$ | **3.13**$_{+2.44}$ |

$^*$fast$_1$ / fast$_{1.5}$: percentage of trajectories containing at least one correct kernel with speedup $> 1$ / $> 1.5$, respectively.

*Table 3.* Ablations on KernelBench and EditBench.

| | KernelBench | | | | | | | | EditBench | |
|---|---|---|---|---|---|---|---|---|---|---|
| | Correctness↑ | | Speedup↑ | | fast$_1$ ↑ | | fast$_{1.5}$ ↑ | | Pass@1 | Pass@16 |
| | best@16 | avg@16 | best@16 | avg@16 | best@16 | avg@16 | best@16 | avg@16 | | |
| *Baseline* | 74 | 12.56 | 1.33 | 1 | 45 | 6.75 | 8 | 0.94 | 53.70 | 55.56 |
| + SEMREP *TTS* | 59 | 15.19 | 1.47 | 1.13 | 28 | 8.19 | 11 | 3.62 | 56.48 | 62.04 |
| + SEMREP-*trained* | 79 | 25.75 | 1.48 | **1.25** | 35 | 13.5 | 11 | **5.75** | 55.56 | 62.96 |
| + *Both* (Ours) | **93** | **37.63** | **2.87** | 1.21 | **72** | **21.06** | **12** | 3.13 | **57.41** | **66.67** |

periments, including baseline finetuning and our proposed SEMREP, to prevent overfitting (detailed in Appendix A.3).

Due to computational budget constraints, we limit our inference to $T = 2$, $b = 1$, and $k = 16$ parallel rollouts at each iteration, though arbitrary $T$, $k$ and $b$ are supported. For the specific case of $T = 2$, the first iteration explores semantically equivalent variants $C_{rep}$ based on the given $C_{src}$, while the second iteration applies instruction-specific transformation to generate $C_{tgt}$ from the selected $C_{rep}$ based on $\mathcal{U}_{sem}$.

### 3.2. Main Results

We compare SEMREP to original models and state-of-the-art RL-finetuned baselines on both general and domain-specific code editing. As shown in Table 1 and Table 2, SEMREP outperforms the baselines across all tasks and models. On KernelBench, SEMREP outperforms the state-of-the-art Kevin-32B (Baronio et al., 2025) by 43.1% in *Correctness best@16* and achieves $1.1\times$ in *Speedup best@16* and *avg@16*. On EditBench, SEMREP with QwQ-32B matches the commercial gpt-o4-mini and significantly outperforms open-weight models (72B) by up to 6.9% in *Pass@1*.

### 3.3. Ablations

We ablate each design component in SEMREP: (1) the generative code representation learning, (2) the TTS that enforces semantics representation exploration before instruction-specific transformation, and (3) their combination. Table 3 shows the results.

Specifically, we consider the following variants to isolate the contribution of each component: (1) *Baseline*: a model finetuned only on instruction-following data combined with the

TTS that directly applies instruction-specific transformation, without incorporating semantics-preserving transformation in either training or inference; (2) +SEMREP *TTS*: the baseline finetuned model combined with SEMREP TTS where explore semantics-preserving transformation is enforced; (3) +SEMREP-*trained*: a model trained with generative code representation learning and then finetuned, combined with the TTS used in the baseline, (4) +*Both*: the SEMREP-trained model combined with SEMREP TTS, representing the full SEMREP approach.

Notably, SEMREP-*trained* model uses *strictly the same training budget* as the *finetuned* baseline, e.g., the combined training iterations for both generative representation learning and finetuning equal to that of the baseline's finetuning iterations, to ensure a fair comparison.

**Generative code representation learning.** Comparing *Baseline* to +SEMREP-*trained* in Table 3, generative code representation learning alone provides consistent improvement across all metrics by 21% in *Speedup avg@16* and $1.05\times$ in *Correctness avg@16* on KernelBench and 3.5% in *Pass@1* and 13.32% in *Pass@16* on EditBench.

**SEMREP TTS.** Comparing *Baseline* to +SEMREP *TTS* in EditBench, SEMREP TTS consistently improve *Pass@1* and *Pass@16* by 5.18% and 10.45% on EditBench, respectively. In KernelBench, SEMREP TTS alone improves 10.5% in *Speedup best@16*. While *Correctness best@16* drops significantly, this can be attributed to the untrained model producing low-quality representations, which leads to incorrect exploration and thus, the wasted computational budget.

**Full SEMREP.** When turning on both training and inference in SEMREP, it outperforms *Baseline* by $2\times$ and $1.2\times$ in *Cor-*

*Table 4.* Comparing SEMREP to Baronio et al. (2025) against unseen hardware devices, i.e., H200, for generalization by measuring performance on KernelBench (Ouyang et al., 2025).

| Model | Correctness↑ best@16 | avg@16 | Speedup↑ best@16 | avg@16 | fast$_1$ ↑ best@16 | avg@16 | fast$_{1.5}$ ↑ best@16 | avg@16 |
|---|---|---|---|---|---|---|---|---|
| Kevin-32B | 56 | 10.81 | 1.51 | 1.14 | 10 | 1.88 | 6 | 1.5 |
| Finetuned QwQ-32B | 61$_{+5}$ | 12.63$_{+1.82}$ | **1.7**$_{+0.19}$ | 1.13$_{+0.01}$ | 15$_{+5}$ | 3.94$_{+2.06}$ | 8$_{+2}$ | 2.08$_{+0.58}$ |
| **SEMREP (Ours)** | **81**$_{+25}$ | **20.38**$_{+9.57}$ | 1.62$_{+0.11}$ | **1.25**$_{+0.11}$ | 17$_{+7}$ | **4.13**$_{+2.25}$ | **8**$_{+2}$ | **2.25**$_{+0.75}$ |

*Table 5.* Comparing SEMREP-trained model to state-of-the-art models against semantics-preserving code transformations.

| Model | Pass@1 ↑ | Consistency ↑ |
|---|---|---|
| QwQ-32B | 47.22 | 76.85 |
| Finetuned QwQ-32B | 50.93$_{+3.71}$ | 75.00$_{-1.85}$ |
| GPT o4-mini | 53.70$_{+6.48}$ | 83.33$_{+6.48}$ |
| **SEMREP (Ours)** | **53.70**$_{+6.48}$ | **88.89**$_{+12.04}$ |

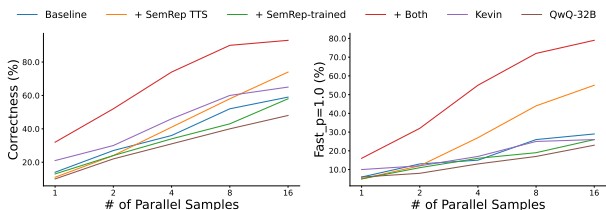

*Figure 3.* Scaling the number of parallel samples on KernelBench.

rectness avg@16 and *Speedup best@16* on KernelBench, and 7% and 20% in *Pass@1* and *Pass@16* on EditBench, respectively. This demonstrates that the training and inference components are designed to work together effectively.

### 3.4. Robustness and Generalization

Real-world code editing must operate reliably across diverse code styles, formats, and syntactic structures. Models that overfit to surface-level syntax will fail when encountering code that deviates from their training distribution. This section evaluates whether SEMREP's explicit training on semantics-preserving transformations improves the model's robustness and generalization to syntactic changes. For example, the SEMREP's transformations could potentially canonicalize the syntactically perturbed code during inference, or help the model reason about the inherent equivalence of the perturbed code via training. To this end, we measure robustness and generalization across two dimensions: (1) robustness to semantics-preserving code perturbations, and (2) generalization to unseen hardware.

**Robustness.** We compare SEMREP to the state-of-the-art baseline against semantics-preserving code transformation. Following ReCode (Wang et al., 2023), we adopt three main types of perturbations: NLAugmenter (Dhole et al., 2023) for docstring, NatGen (Chakraborty et al., 2022) for code syntax, and code format. However, KernelBench input mainly consists of PyTorch API calls, leaving limited room for these perturbations. Therefore, we evaluate robustness only on EditBench.

As shown in Table 5, we report (1) *Pass@1* after the testing samples are perturbed, and (2) *Consistency*, which measures the proportion of samples have maintained their results after the perturbation, i.e., the pass/fail outcome of each test cases is preserved. SEMREP remains robust and less suscep-

tible than *Baseline*, achieving 6.7% better consistency and remains the highest *Pass@1* after perturbations.

**Cross-device generalization.** Low-level CUDA optimizations (e.g., shared memory tiling, warp-level primitives) are highly sensitive to hardware architecture. We investigate SEMREP's generalizability to unseen devices when performing low-level domain-specific CUDA optimization. To evaluate whether the trained models overfit to the training hardware, we evaluate on unseen H200 devices without any additional training. Table 4 demonstrates that SEMREP generalizes effectively to unseen devices, outperforming *Baseline* (Finetuned QwQ-32B) by 10.6% in *Speedup avg@16* while maintaining 61.4% higher in *Correctness avg@16*.

### 3.5. Test Time Scaling

As SEMREP produces explicit code representations as semantics-preserving programs during inference, it naturally supports test-time scaling, where additional compute is used to balance exploration (exploring various equivalent code implementations) and exploitation (sticking to narrow but effective transformations). For example, AlphaEvolve (Novikov et al., 2025) scales search by exploring semantically equivalent implementations of the same algorithm, while prior program repair works like REx (Tang et al., 2024) also formulate the iterative program refinement as explore–exploit trade-offs. By exposing semantic structure rather than internalizing it in weights, SEMREP's output serves as meaningful stepping stones for later search.

To evaluate the amenability of SEMREP to test-time scaling, we study how *best@k* performance scales when increasing the number of parallel trajectories $k$, while fixing the number of refinement turns. We compare against the base QwQ-32B, the state-of-the-art Kevin-32B (Baronio et al., 2025),

**(a) Direct transformation.**

```
Step 1. Applying "run these in parallel":
@staticmethod
async def generate_responses_for_models(prompt: str, models: List[str], **kwargs:
Any) -> List[str]:
    responses = []
    providers = []
    for model in models:
        provider_for_model = LLMArena._get_provider_for_model(model, **kwargs)
        providers.append(provider_for_model)
-       for provider in providers:
-           try:
-               response = await provider.generate_chat_response(prompt)
-               responses.append(response)
-           except Exception as e:
-               logging.error(f"Error generating response from {provider}: {e}")
-               responses.append(f"Error generating response from {provider}: {e}")
+       tasks = [provider.generate_chat_response(prompt) for provider in providers]
+       for task in asyncio.as_completed(tasks):
+           try:
+               response = await task
+               responses.append(response)
+           except Exception as e:
+               provider = providers[tasks.index(task)]
+               logging.error(f"Error generating response from {provider}: {e}")
+               responses.append(f"Error generating response from {provider}: {e}")
    return responses

Step 2. Resolving missing import
+import asyncio
@staticmethod
async def generate_responses_for_models(...)
```

**(b) SEMREP.**

```
Step 1. Semantics-preserving transformation
+import asyncio
@staticmethod
async def generate_responses_for_models(prompt: str, models: List[str], **kwargs: Any)
-> List[str]:
    responses = []
    providers = []
    for model in models:
        provider_for_model = LLMArena._get_provider_for_model(model, **kwargs)
        providers.append(provider_for_model)
-       for provider in providers:
-           try:
-               response = await provider.generate_chat_response(prompt)
-               responses.append(response)
-           except Exception as e:
-               logging.error(f"Error generating response from {provider}: {e}")
-               responses.append(f"Error generating response from {provider}: {e}")
+       async def generate_with_catch(provider, prompt):
+           try:
+               return await provider.generate_chat_response(prompt)
+           except Exception as e:
+               msg = f"Error generating response from {provider}: {e}"
+               logging.error(msg)
+               return msg
+       responses = [await generate_with_catch(provider, prompt) for provider in providers]
    return responses

Step 2. Applying "run these in parallel":
-       responses = [await generate_with_catch(provider, prompt) for provider in providers]
+       tasks = [generate_with_catch(provider) for provider in providers]
+       responses = await asyncio.gather(*tasks)
```

*Figure 4.* An example in EditBench showing how (a) direct transformation and (b) SEMREP generate with "run these in parallel".

and all variants from Section 3.3.

As shown in Figure 3, SEMREP grows faster than all the baselines and variants when the number of parallel $k$ increases. At $k = 16$, SEMREP achieves up to $3\times$ gains over *Baseline* (Finetuned QwQ-32B) in *fast*$_{1.0}$ *best@16*, while maintaining $4\times$ higher in *Correctness best@16*.

### 3.6. Case Study on EditBench

As shown in Figure 4, we examine how direct transformation and SEMREP handle a real-world code editing task from EditBench that requires implementing parallelism. Full trajectories are presented in Appendix C. The original code (see Listing 10 for the complete version) takes a prompt and a list of LLM models, calls each model's API sequentially via `generate_chat_response`, and returns a list of outputs in the same order as the input models. Each call must complete before the next begins, so the total execution time is the sum of per-model response times.

**Direct transformation.** Direct transformation (Figure 4a) conflates rewriting and optimization at each step, which makes it harder to get both right simultaneously. Therefore, in *Step 1*, it attempts to introduce parallelism by converting the sequential loop into concurrent API calls using `asyncio.as_completed`, an async API that returns results as soon as each call finishes. However, it omits the required import for `asyncio`. Upon the external execution feedback, *Step 2* attempts to fix this compilation error rather than making any meaningful progress. While *Step 2* fixes the issue, it breaks the semantics of output ordering even though the code runs faster. The original code appends outputs in the order they are received as input, whereas `asyncio.as_completed` returns results in completion order rather than the same order as the input models. As a result, the $i$-th output list element no longer reliably matches the $i$-th model in the input list.

**SEMREP.** SEMREP (Figure 4b) decouples the semantic-preserving refactoring required for the optimization from the optimization itself by separating them into two distinct steps. In *Step 1*, SEMREP is prompted to produce a semantic equivalent but not necessarily optimized code. As a result, SEMREP shuffles code around and extracts a helper function `generate_with_catch` to collect the API responses. Note that this transformation does not optimize the performance but abstracts the per-call logic into a self-contained module, making the parallelism opportunity more explicit in structure. In *Step 2*, SEMREP is prompted to follow the user's instructions to optimize the refactored code from *Step 1*. With the helper function in place, SEMREP can focus on implementing parallel execution of the API calls. Specifically, SEMREP uses the async API `asyncio.gather`, which concurrently runs the helper function for all models and returns the responses in the same order as the input models. By separating the required semantics-preserving refactoring (*Step 1*) from the instruction following (*Step 2*), SEMREP avoids the correctness pitfalls that potentially arise when both goals compete within a single generation.

### 3.7. Integrating SEMREP with Evolutionary Agent

As SEMREP is explicitly trained on semantics-equivalent transformations, it has the potential to produce diverse intermediate programs not yet optimized, but potentially include rewrites helpful to enable future optimizations. This behavior aligns well with evolutionary search, which also benefits from exploring non-greedy intermediate states. Therefore, we embed SEMREP into the evolutionary agentic framework OpenEvolve (Sharma, 2025) for code optimizations. All compared models, i.e., DeepSeek-V3-Chat and DeepSeek-V3-Reasoner, are run within OpenEvolve under the same agent configurations. See Appendix B for implementation details.

**Minimum spanning tree (MST).** We adopt the minimum

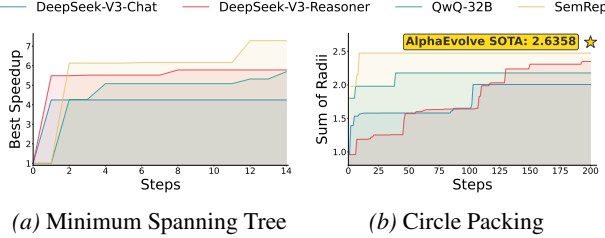

*(a)* Minimum Spanning Tree          *(b)* Circle Packing

*Figure 5.* Best results over optimization steps with OpenEvolve for the minimum spanning tree and circle packing problem.

spanning tree task from AlgoTune (Press et al., 2026) for the case study. This problem seeks to find a subset of edges that connects all vertices in a weighted, undirected graph with the lowest possible total edge weight and no cycles.

As shown in Figure 5a, SEMREP achieves a speedup of 7.28x, outperforming both 685B DeepSeek-V3-Chat and DeepSeek-V3-Reasoner (Liu et al., 2024a) by 25.95% at step 14. Starting from the initial NetworkX (Hagberg et al., 2007) implementation (Listing 3), besides replacing NetworkX with a custom Kruskal's algorithm, SEMREP achieves a significant $6.13\times$ speedup through implementing the Union-Find algorithm in an iterative way, which all the other models fail to identify (see Listing 1 and 2). Ultimately, SEMREP achieves the final $7.28\times$ speedup by applying micro-optimizations, including `operator.itemgetter` for sorting and precomputation to improve cache locality (Listing 6). See Appendix B.1 for detailed analysis.

```
1  def find(u: int) -> int:
2      while parent[u] != u:
3          # Path compression
4          parent[u] = parent[parent[u]]
5          u = parent[u]
6      return u
```

*Listing 1.* SEMREP implements the Union-Find algorithm with efficient while loop.

```
1  def _find(self, parent: List[int], x: int) -> int:
2      """Find with path compression."""
3      if parent[x] != x:
4          parent[x] = self._find(parent, parent[x])
5      return parent[x]
```

*Listing 2.* All the other models implement Union-Find with recursion.

**Circle packing.** We adopt the circle packing problem studied in AlphaEvolve (Novikov et al., 2025). The circle packing problem is a constrained optimization task that seeks to arrange $n = 26$ non-overlapping circles within a unit square to maximize the sum of their radii.

As shown in Figure 5b, integrating SEMREP into OpenEvolve achieves the sum radii of 2.4737 (93.88% of the state-of-the-art AlphaEvolve results). Notably, AlphaEvolve leverages Gemini models over many thousands of iterations,

while our SEMREP achieves competitive performance under substantially more constrained resources. Moreover, it still significantly outperforms OpenEvolve paired with 685B DeepSeek-V3-Chat and Reasoner models. Through evolutionary search, SEMREP discovers an effective grid-based pattern in the early stage, and then switches to an alternative ring-based approach that temporarily reduces the performance. Although this semantics-preserving change does not yield immediate improvement and even harms the performance, the algorithmic shift opens up new opportunities for further optimization. Ultimately, at step 93, SEMREP achieves the improvement by starting from a ring-based initialization that provides a favorable prior, and then applies the simulated annealing to relocate components for better performance. See Appendix B.2 for details.

# 4. Discussion and Limitations

**Test-based equivalence.** According to Rice's theorem, the general semantic equivalence of two arbitrary programs is fundamentally not decidable. SEMREP thus adopted EMI to formalize the approximate semantic equivalence. This is subject to the availability of tests for computing verifiable rewards during representation learning, and lacks a soundness guarantee. If these tests have low coverage, i.e., the inputs do not exercise most of the code, the SEMREP-trained model may generate biased code that is only semantically equivalent on a small part of the code logic. A quick investigation of our training samples confirmed that the tests used during mid-training achieved ≥99% line coverage. Our investigation also shows that relaxed coverage permits more exploratory, but potentially semantics-breaking edits, while stricter coverage enforces tighter behavioral consistency (see Section F). This serves as an exciting future work.

**Reward hacking.** During the generative code representation learning, we reject exact duplicates to prevent reward hacking, where a model could trivially return the input code unchanged to maximize the reward. However, minor surface-level refactorings, e.g., adding comments, renaming variables, are deliberately allowed. These lightweight transformations may help the model understand code behavior, building a richer representation that can, in turn, inspire more high-quality transformations at inference, such as performance optimization and functional adaptation to new requirements. Determining which semantic-preserving transformations are most beneficial for model training remains an open question. It is also challenging to directly measure how the explicit code representation learning encourages more exploration and leads to more creative transformations, without resorting to the outcomes from the downstream editing tasks. We are working on including additional categories of useful semantics-preserving transformations beyond those presented in this paper.

**Similarity to curriculum learning.** Our scheduling of the reward for representation learning and finetuning can also be made continuous as a curriculum, i.e., start from a high reward on generating semantically equivalent code, and gradually decrease it to prioritize instruction-specific transformation. The key advantage of SEMREP's strict two-stage training is that its mid-training is a one-time effort that facilitates many downstream editing tasks, rather than being specialized in a single task-specific setting, which can suffer from limited generalization.

**Evolutionary agentic framework.** While SEMREP utilizes an iterative inference paradigm, the current implementation is largely simplified due to computational constraints. This setup may not fully unleash the potential of the SEMREP's trained model for challenging coding tasks that require many steps to solve. While our case studies on integrating SEMREP with evolutionary coding agents demonstrate promising results, we leave a systematic study and broader application to real-world projects for future work.

## 5. Related Work

**Code editing and transformation.** Code editing has emerged as a critical application of LLMs in software engineering, with applications spanning code maintenance (Chi et al., 2025; Jimenez et al., 2024; Wei et al., 2026), performance optimization (Shypula et al., 2023; Huang et al., 2024; Peng et al., 2025; Garg et al., 2022; Press et al., 2026; Shetty et al., 2026), bug fixing (Xia & Zhang, 2024; Jin et al., 2023), code migrations (Liu et al., 2025; Zhang et al., 2025). Most existing approaches either adopt finetuning to learn a direct mapping between code pairs or leverage iterative refinement with execution or self-generated feedback (Huang et al., 2024; Peng et al., 2025; Huang et al., 2023; Xia & Zhang, 2024; Chen et al., 2024; Dong et al., 2024; Madaan et al., 2024; Zelikman et al., 2024; Liu et al., 2024b). While these methods have shown effectiveness, they treat the semantic understanding as an implicit byproduct of the editing task rather than an explicit learning objective. In contrast, SEMREP complements these approaches by treating semantics-preserving transformation as an explicit foundational capability, explicitly grounding transformation reasoning in verifiable representations that benefit both training and inference.

**Code semantics learning.** Previous works have attempted to capture code semantics through execution-aware learning (Pei et al., 2020; Ni et al., 2024; Liu et al., 2023; Ding et al., 2024) or structural constraints like program dependence graphs (Pei et al., 2023; Guo et al., 2020). However, these methods are limited to specific program representations with nontrivial construction costs and limited generality. SEMREP formulates semantic understanding as a gener-ative task where the model explicitly produces semantically equivalent code as a verifiable intermediate representation, making the semantic understanding amenable to reinforcement learning.

**Reinforcement pre-training and evolutionary search.** SEMREP shares a similar spirit to reinforcement pre-training (Hatamizadeh et al., 2025; Dong et al., 2025; Copet et al., 2025), where the exploration is encouraged and introduced during pre-training before task-specific training to improve generalization. SEMREP also relates to evolutionary agentic frameworks (Sharma, 2025; Novikov et al., 2025; Tang et al., 2024), where inference is designed to balance exploration and exploitation (Tang et al., 2024). SEMREP extends the idea by explicitly disentangling the representation learning from instruction-specific transformation, enabling LLMs to leverage improved semantic understanding for improved performance and generalization (Section 3.4).

## 6. Conclusion

We introduce SEMREP, a code transformation framework that decouples semantic understanding from code editing. Our key approach is to employ *generative code representation learning* to enable models to reason about program behavior as explicit representations. SEMREP enables smaller models to match or exceed larger models, achieving significant gains across both general and domain-specific code editing tasks, while exhibiting enhanced robustness to semantics-preserving transformations and improved generalization to unseen hardware.

## Acknowledgement

We thank Chenghao Yang and Zhicheng Zhang for valuable feedback on reinforcement learning training, and Mingyuan Xiang for insightful discussions and help with kernel performance analysis. We are also grateful to the anonymous reviewers for their constructive comments and feedback, which significantly improved the quality of this paper. We are also grateful to Jun Yang for his assistance with parts of the experimental setup. This work was supported in part by Chameleon (Keahey et al., 2020) and OpenAI Researcher Access Program (OpenAI).

## Impact Statement

Code editing is a fundamental component of the modern developer workflow. While large language models (LLMs) have demonstrated promise in automated code transformation, their end-to-end nature can occasionally miss subtle semantic cues, leading to silent logic errors and overlooked optimization opportunities. Our paper introduced a new ap-

proach to enhance LLMs' understanding of code semantics under the same budget of finetuning-only methods, enabling more effective and reliable code transformations. Such capability can significantly impact the development of safe, efficient, and trustworthy software.

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

# A. Experiment Details

## A.1. Prompt Details

---

**Prompt 1: Prompt for semantic-preserving transformation on KernelBench.**

You are given the following architecture:
`{ref_arch_src}`

Perform a semantic-preserving transformation on the given architecture. This means you should rewrite the code in a way that:
- Retains the exact behavior and output of the original code
- Changes the style, structure, algorithm, or abstraction level
- Improves the code structure without changing its semantics
- You are free to be creative with the transformation while ensuring semantic equivalence

Use `torch.utils.cpp_extension.load_inline` and name your optimized output architecture `ModelNew`. Output the new code in code blocks. Please generate real code, NOT pseudocode, and make sure the code compiles and is fully functional. After your answer, summarize your changes in a few sentences.

---

**Prompt 2: Prompt for instruction-specific transformation on KernelBench.**

You are given the following architecture:
`{ref_arch_src}`

Optimize the architecture named `Model` with custom CUDA operators! Use techniques like:
- Shared memory for tile-based computation
- Coalesced memory access
- Warp-level primitives
- Tensor cores if applicable

Use `torch.utils.cpp_extension.load_inline` and name your optimized output architecture `ModelNew`. Output the new code in code blocks. Please generate real code, NOT pseudocode, and make sure the code compiles and is fully functional. After your answer, summarize your changes in a few sentences.

---

**Prompt 3: Prompt for semantic-preserving transformation on EditBench.**

You are given the following code to modify:
The Original code (to be modified):

`{lang}`
`{original_code}`

Perform a semantic-preserving transformation on the code that:
- Maintains the exact behavior and output for the rest of the code
- Uses semantic-preserving transformations where possible (e.g., restructuring, renaming variables, using equivalent operations)
- Preserves all side effects, error behaviors, and corner-case handling

Please output the entire modified code file in a code block beginning with "```{lang}```". After your answer, summarize your changes in a few sentences.

---

---

Prompt 4: Prompt for instruction-specific transformation on EditBench.

Generate a new implementation of the following code based on the user instruction:

The Original code (to be modified):

```
{lang}
{original_code}
```

The user instruction is:
```
{instruction}
```

And they highlighted this section to be changed:
```
{lang}
{highlighted_code}
```

Please only change the highlighted section and leave the rest of the code unchanged.
Please output the entire code file.
Respond only in a code block beginning with ```{lang}```.

---

## A.2. Dataset and Hyperparameters

**Dataset.** For EditBench, we train on 82 randomly selected CodeContests samples (Li et al., 2022; Shypula et al., 2023) and the EDIT-Bench-complete dataset (excluding the core test set) and evaluate on the core set. Since EditBench does not distinguish between tests for preserved functionality and intended new functionality, we manually run the provided tests on the input code. Tests that the original code can pass are considered as semantic preservation verification ($\mathcal{U}_{sem}$), while tests that the original code fails to pass (but the expected edited code should pass) are treated as instruction adherence verification ($\mathcal{U}_{edit}$).

For KernelBench, following Kevin (Baronio et al., 2025), we randomly select 180 tasks from the benchmark for training and use the remaining 100 for testing. During evaluation, we also identified and fixed a bug in the KernelBench speedup calculation. Specifically, the original implementation did not enforce consistent sorting by *problem_id* before performing pair-wise speedup comparisons. Thus, when evaluation results were incomplete, this inconsistency led to mismatched pairs and statistically incorrect speedup ratios.

**Hyperparameters.** We set the maximum training sequence length to 16,384 tokens to accommodate long reasoning trajectories. The learning rate is set to 2e-6, and the batch size is 8. We set $\alpha_1 = 1.3, \beta_1 = 0.5$ for generative code representation learning, and $\alpha_2 = 1, \beta_2 = 0.3, \gamma = 0.1$ for instruction-specific transformation learning. During inference, we set the maximum sequence length to 16,384 tokens, with temperature 0.7 and top_p=0.9 to balance exploration and exploitation. We use $\omega_1 = 1, \omega_2 = 0$ for semantics-preserving transformation, and $\omega_1 = 0.3, \omega_2 = 0.7$ for instruction-following transformation.

## A.3. Budget

In order to achieve a fair comparison, we make sure the SEMREP-trained model uses strictly the same training budget compared to the extensively finetuned baselines. We consider the total training budget $\mathcal{B}$ as the total number of training steps used by *Baseline*. For SEMREP, we take $\mathcal{B}/2$ steps for representation learning, and the other $\mathcal{B}/2$ steps for instruction-specific learning.

## A.4. Things We Tried that Did not Work

We experimented with several alternative ways to learn code representations, but found that they underperformed.

**Separate samples for distinct learning phases.** Generative code representation learning and instruction-specific transformation learning should be conducted on different data samples to prevent conflicting learning signals. Mixing or overlapping

the same instances across both phases led to gradient conflicts and degraded performance.

**Stable learning objectives.** The learning objective (generative code representation vs. instruction-specific transformation reward) should not be switched too rapidly during training. Frequently alternating objectives caused the evolving rewards problem, where the stale training experiences become misleading, resulting in degraded performance.

# B. OpenEvolve Case Study

This section provides implementation details for integrating our SEMREP-trained QwQ-32B into OpenEvolve (Sharma, 2025). We demonstrate the integration using two case studies: the minimum spanning tree task from AlgoTune (Press et al., 2026) and the circle packing problem.

## B.1. AlgoTune

**Hyperparameters.** The evolutionary search was configured with a population size of 1000 and an archive size of 100, distributed across 4 islands to promote diversity. To balance search dynamics, we set the elite selection ratio to 0.1, exploration ratio to 0.3, and exploitation ratio to 0.6. For island communication, we employed a migration rate of 0.1. During prompt construction, we included the top 3 programs and 2 diverse programs. The evolution was run for a maximum of 16 iterations. The LLM was configured with a temperature of 0.7, top-$p$ of 0.9, and a maximum generation length of 16,384. The `random_seed` was 42, and `diff_based_evolution` was set to false.

**Evaluator modifications.** We refined the evaluation metric to better align with the goal of performance optimization. The original AlgoTune evaluator calculated a composite score as follows:

$$\text{score} = 0.7 \cdot \text{Correctness} + 0.2 \cdot \text{Performance} + 0.1 \cdot \text{Reliability} \tag{1}$$

In this formula, *Reliability* measures the ratio of successful executions to total trials. *Correctness* represents the proportion of valid solutions among successful runs, and *Performance* is calculated as $1/(1 + t)$, where $t$ denotes the execution time in milliseconds. We found that the absolute runtime dependency in the performance term introduced variance due to baseline fluctuations, making it an unstable proxy. Consequently, we modified the evaluator (e.g., in `minimum_spanning_tree/evaluator.py`) to define the fitness score directly as the *speedup* ratio relative to the baseline implementation for correct solutions, while assigning a score of zero if incorrect. This ensures that the evolutionary process directly optimizes for relative efficiency gains unaffected by environmental noise. All compared methods were re-evaluated under this modified evaluator.

**Trajectory analysis.** Starting from the initial NetworkX implementation, SEMREP evolves to achieve a 7.28× speedup, outperforming both QwQ-32B and DeepSeek-V3-Reasoner. We track the evolution trajectories of SEMREP in detail, examining important breakthroughs at each stage of its evolution:

- **Initial implementation (Listing 3).** The initial implementation uses the NetworkX library (Hagberg et al., 2007) for graph construction and MST computation. The implementation suffers from significant library overhead, complex graph data structures, and multiple data format conversions.

- **Custom algorithm implementation (Listing 4).** SEMREP replaces the NetworkX-based implementation with a custom MST implementation. The key changes that lead to the performance breakthrough include: (1) elimination of NetworkX dependency and library overhead, (2) direct implementation of Kruskal's algorithm with greedy edge selection, (3) an iterative Union-Find data structure, and (4) early exit optimization once the MST is complete. This implementation represents a 6.13× speedup over the baseline, demonstrating SEMREP's ability to recognize performance bottlenecks and discover efficient algorithms.

- **Stabilization (Listing 5).** SEMREP maintains the optimized algorithm while improving code quality with type hints and better documentation. Performance remains stable at around 6× speedup.

- **Final micro-optimizations (Listing 6).** SEMREP applies micro-optimizations that improve cache locality and reduce function call overhead, achieving the final 7.28× speedup. The key optimizations include: (1) using `operator.itemgetter` instead of lambda functions for sorting, (2) precomputing the target value (num_nodes

- 1) to avoid repeated calculations in the tight loop, and (3) consistent use of C-implemented operator functions for all sorting operations. These micro-optimizations provide an additional 21% improvement.

```python
1  import logging
2  import random
3  import networkx as nx
4  import numpy as np
5  from typing import Any, Dict, List, Optional
6
7  class MinimumSpanningTree:
8      """
9      Initial implementation of minimum_spanning_tree task.
10     This will be evolved by OpenEvolve to improve performance and correctness.
11     """
12
13     def __init__(self):
14         """Initialize the MinimumSpanningTree."""
15         pass
16
17     def solve(self, problem):
18         """
19         Solve the minimum_spanning_tree problem.
20
21         Args:
22             problem: Dictionary containing problem data specific to
    minimum_spanning_tree
23
24         Returns:
25             The solution in the format expected by the task
26         """
27         try:
28             """
29             Construct the graph in networkx, compute MST using
    minimum_spanning_edges,
30             and return the MST edges sorted by (u, v).
31
32             :param problem: dict with 'num_nodes', 'edges'
33             :return: dict with 'mst_edges'
34             """
35             G = nx.Graph()
36             num_nodes = problem["num_nodes"]
37             edges = problem["edges"]
38
39             G.add_nodes_from(range(num_nodes))
40             for u, v, w in edges:
41                 G.add_edge(u, v, weight=w)
42
43             # networkx returns an iterator of MST edges
44             mst_edges_data = list(nx.minimum_spanning_edges(G, data=True))
45
46             # Convert to [u, v, weight]
47             # each edge is (u, v, {'weight': w})
48             mst_edges = []
49             for u, v, data in mst_edges_data:
50                 # ensure u < v for sorting consistency
51                 if u > v:
52                     u, v = v, u
53                 mst_edges.append([u, v, data["weight"]])
54
55             # Sort by (u, v)
56             mst_edges.sort(key=lambda x: (x[0], x[1]))
57             return {"mst_edges": mst_edges}
58
59         except Exception as e:
```

```
60              logging.error(f"Error in solve method: {e}")
61              raise e
```

*Listing 3.* MST: initial implementation using NetworkX.

```python
1  import logging
2  from typing import Any, Dict, List, Optional
3
4  class MinimumSpanningTree:
5      """
6      Implementation of minimum_spanning_tree task using Kruskal's algorithm for
       improved performance.
7      """
8
9      def __init__(self):
10         """Initialize the MinimumSpanningTree."""
11         pass
12
13     def solve(self, problem: Dict[str, Any]) -> Dict[str, List[List[float]]]:
14         """
15         Solve the minimum_spanning_tree problem using Kruskal's algorithm.
16
17         Args:
18             problem: Dictionary containing problem data specific to
       minimum_spanning_tree
19
20         Returns:
21             The solution in the format expected by the task
22         """
23         try:
24             num_nodes = problem["num_nodes"]
25             edges = problem["edges"]
26
27             # Sort edges by weight to facilitate Kruskal's algorithm
28             edges_sorted = sorted(edges, key=lambda x: x[2])
29
30             parent = list(range(num_nodes))
31             size = [1] * num_nodes  # Using size for union by larger size
32
33             def find(u: int) -> int:
34                 while parent[u] != u:
35                     parent[u] = parent[parent[u]]  # Path compression
36                     u = parent[u]
37                 return u
38
39             def union(u: int, v: int) -> bool:
40                 u_root = find(u)
41                 v_root = find(v)
42                 if u_root == v_root:
43                     return False  # Already connected
44                 # Union by size: attach smaller to larger
45                 if size[u_root] < size[v_root]:
46                     parent[u_root] = v_root
47                     size[v_root] += size[u_root]
48                 else:
49                     parent[v_root] = u_root
50                     size[u_root] += size[v_root]
51                 return True
52
53             mst_edges = []
54             edge_count = 0
55
56             for edge in edges_sorted:
57                 u, v, w = edge
58                 if find(u) != find(v):
```

```
59                    union(u, v)
60                    # Ensure consistent ordering for output sorting
61                    if u > v:
62                        u, v = v, u
63                    mst_edges.append([u, v, w])
64                    edge_count += 1
65                    if edge_count == num_nodes - 1:
66                        break  # Early exit once MST is complete
67
68            # Final sort by (u, v) for consistent output
69            mst_edges.sort(key=lambda x: (x[0], x[1]))
70            return {"mst_edges": mst_edges}
71
72        except Exception as e:
73            logging.error(f"Error in solve method: {e}")
74            raise e
```

*Listing 4.* MST: custom Kruskal's algorithm with Union-Find.

```
1  import logging
2  from typing import Any, Dict, List, Optional
3
4  class MinimumSpanningTree:
5      """
6      Implementation of minimum_spanning_tree task using Kruskal's algorithm for
       improved performance.
7      """
8
9      def __init__(self):
10         """Initialize the MinimumSpanningTree."""
11         pass
12
13     def solve(self, problem: Dict[str, Any]) -> Dict[str, List[List[float]]]:
14         """
15         Solve the minimum_spanning_tree problem using Kruskal's algorithm.
16
17         Args:
18             problem: Dictionary containing problem data specific to
       minimum_spanning_tree
19
20         Returns:
21             The solution in the format expected by the task
22         """
23         try:
24             num_nodes = problem["num_nodes"]
25             edges = problem["edges"]
26
27             # Sort edges by weight to facilitate Kruskal's algorithm
28             edges_sorted = sorted(edges, key=lambda x: x[2])
29
30             parent = list(range(num_nodes))
31             size = [1] * num_nodes  # Using size for union by larger size
32
33             def find(u: int) -> int:
34                 while parent[u] != u:
35                     parent[u] = parent[parent[u]]  # Path compression
36                     u = parent[u]
37                 return u
38
39             def union(u: int, v: int) -> bool:
40                 u_root = find(u)
41                 v_root = find(v)
42                 if u_root == v_root:
43                     return False  # Already connected
44                 # Union by size: attach smaller to larger
```

```
45                    if size[u_root] < size[v_root]:
46                        parent[u_root] = v_root
47                        size[v_root] += size[u_root]
48                    else:
49                        parent[v_root] = u_root
50                        size[u_root] += size[v_root]
51                    return True
52
53            mst_edges = []
54            edge_count = 0
55
56            for edge in edges_sorted:
57                u, v, w = edge
58                if find(u) != find(v):
59                    union(u, v)
60                    # Ensure consistent ordering for output sorting
61                    if u > v:
62                        u, v = v, u
63                    mst_edges.append([u, v, w])
64                    edge_count += 1
65                    if edge_count == num_nodes - 1:
66                        break  # Early exit once MST is complete
67
68            # Final sort by (u, v) for consistent output
69            mst_edges.sort(key=lambda x: (x[0], x[1]))
70            return {"mst_edges": mst_edges}
71
72        except Exception as e:
73            logging.error(f"Error in solve method: {e}")
74            raise e
```

*Listing 5.* MST: code quality improvements.

```
1 import logging
2 import operator
3 from typing import Any, Dict, List, Optional
4
5 class MinimumSpanningTree:
6     """
7     Implementation of minimum_spanning_tree task using Kruskal's algorithm for
      improved performance.
8     """
9
10    def __init__(self):
11        pass  # No initialization needed
12
13    def solve(self, problem: Dict[str, Any]) -> Dict[str, List[List[float]]]:
14        try:
15            num_nodes = problem["num_nodes"]
16            edges = problem["edges"]
17
18            # Sort edges by weight using efficient operator.itemgetter
19            edges_sorted = sorted(edges, key=operator.itemgetter(2))
20
21            parent = list(range(num_nodes))
22            size = [1] * num_nodes
23
24            def find(u: int) -> int:
25                while parent[u] != u:
26                    parent[u] = parent[parent[u]]  # Path compression
27                    u = parent[u]
28                return u
29
30            def union(u: int, v: int) -> bool:
31                u_root = find(u)
```

```
32                  v_root = find(v)
33                  if u_root == v_root:
34                      return False
35                  if size[u_root] < size[v_root]:
36                      parent[u_root] = v_root
37                      size[v_root] += size[u_root]
38                  else:
39                      parent[v_root] = u_root
40                      size[u_root] += size[v_root]
41                  return True
42
43              mst_edges = []
44              edge_count = 0
45              target = num_nodes - 1  # Precompute to avoid repeated calculation
46
47              for edge in edges_sorted:
48                  u, v, w = edge
49                  if find(u) != find(v):
50                      union(u, v)
51                      if u > v:
52                          u, v = v, u
53                      mst_edges.append([u, v, w])
54                      edge_count += 1
55                      if edge_count == target:
56                          break
57
58              # Final sort using operator for efficiency
59              mst_edges.sort(key=operator.itemgetter(0, 1))
60              return {"mst_edges": mst_edges}
61
62          except Exception as e:
63              logging.error(f"Error in solve method: {e}")
64              raise e
```

*Listing 6.* MST: micro-optimized implementation.

## B.2. Circle Packing

**Hyperparameters.** We employ a two-phase evolutionary search starting from our baseline. The LLM configuration is the same as AlgoTune. Phase 1 uses a population size of 60, an archive size of 25, 4 islands, an elite selection ratio 0.3, an exploitation ratio 0.7, the top 3 programs in prompts, and runs for 100 iterations. Phase 2 uses a population size of 70, an archive size of 30, 5 islands, an elite selection ratio 0.3, an exploitation ratio 0.6, the top 4 programs in prompts, and runs for 100 iterations.

**Trajectory analysis.** The state-of-the-art result for this circle packing problem is 2.6358, achieved by AlphaEvolve (Novikov et al., 2025). We track the evolution trajectories of SEMREP in detail, examining important breakthroughs at each stage of its evolution:

- **Early discovery (Listing 7).** SEMREP discovers an effective grid-based pattern achieving 2.4736 (93.88% of the target) at iteration 9. Specifically, SEMREP uses a systematic grid placement with the 26th circle at the true center (0.5, 0.5) and improved radius computation using edge distances and half the closest neighbor distance.

- **Exploration (Listing 8).** SEMREP abandons the grid pattern and explores a ring-based approach, which reduces the performance to 2.036 (77.27% of the target). This demonstrates that SEMREP actively explores diverse semantic-preserving representations that might even lead to temporary performance degradation.

- **Final optimization (Listing 9).** Building on the ring-based representation discovered during exploration, SEMREP achieves the highest score of 2.4737 through extensive simulated annealing, surpassing the previous peak. This final outcome is consistent with our intuition that a temporary degradation in performance might expose additional optimization opportunities and ultimately lead to superior outcomes.

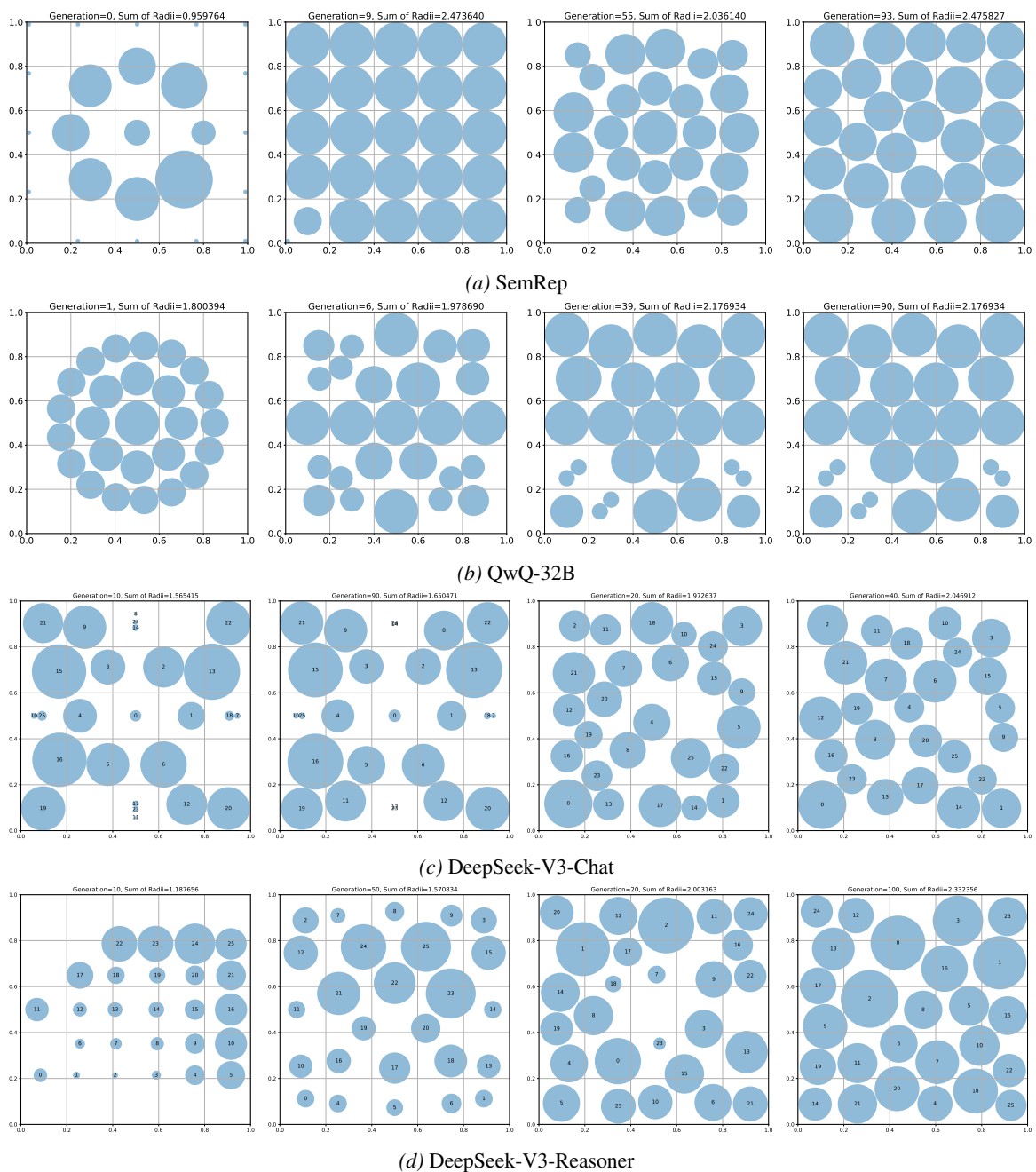

*Figure 6.* Visualized trajectories of circle packing optimization, illustrating the iterative refinement of configurations across different models.

We also illustrate the evolution trajectories for all models in Figure 6, offering a high-level view of how each model progressively refines circle arrangements through iterative optimization.

```python
"""Constructor-based circle packing for n=26 circles"""
import numpy as np

def construct_packing():
    """
    Construct a specific arrangement of 26 circles in a unit square
    that attempts to maximize the sum of their radii.

```

```python
 9       Returns:
10           Tuple of (centers, radii, sum_of_radii)
11           centers: np.array of shape (26, 2) with (x, y) coordinates
12           radii: np.array of shape (26) with radius of each circle
13           sum_of_radii: Sum of all radii
14       """
15       n = 26
16       centers = np.zeros((n, 2))
17
18       # Create a 5x5 grid with spacing 0.2, skipping the central cell
19       spacing = 0.2
20       idx = 0
21       for i in range(5):
22           y = 0.1 + i * spacing
23           for j in range(5):
24               x = 0.1 + j * spacing
25               if i == 2 and j == 2:  # Skip the center cell (0.5, 0.5)
26                   continue
27               centers[idx] = [x, y]
28               idx += 1
29
30       # Add the 26th circle at the true center
31       centers[25] = [0.5, 0.5]
32
33       # Clip to ensure all circles are within the unit square
34       centers = np.clip(centers, 0.01, 0.99)
35
36       # Compute radii using the improved method
37       radii = compute_max_radii(centers)
38       sum_radii = np.sum(radii)
39
40       return centers, radii, sum_radii
41
42 def compute_max_radii(centers):
43       """
44       Compute the maximum valid radii for each circle position without overlapping or
         exceeding edges.
45       Uses edge distances and half the closest neighbor distance.
46
47       Args:
48           centers: np.array of shape (n, 2) with (x, y) coordinates
49
50       Returns:
51           np.array of shape (n) with radius of each circle
52       """
53       n = centers.shape[0]
54       radii = np.empty(n)
55
56       # Compute edge distances for each circle
57       edge_dists = np.minimum(centers[:, 0], 1 - centers[:, 0])
58       edge_dists = np.minimum(edge_dists, centers[:, 1])
59       edge_dists = np.minimum(edge_dists, 1 - centers[:, 1])
60
61       # Compute minimum distance to any other circle divided by 2
62       min_dists = np.full(n, np.inf)
63       for i in range(n):
64           for j in range(i + 1, n):
65               dist = np.linalg.norm(centers[i] - centers[j])
66               if dist < min_dists[i]:
67                   min_dists[i] = dist
68               if dist < min_dists[j]:
69                   min_dists[j] = dist
70       min_dists /= 2.0  # Max possible radius without overlapping closest neighbor
71
72       # The radius is the minimum of edge distance and min neighbor distance/2
```

```
73        radii = np.minimum(edge_dists, min_dists)
74
75        return radii
76
77 # EVOLVE-BLOCK-END
```

*Listing 7.* Circle packing implementation at iteration 9: grid-based pattern with improved radius computation.

```
1 import numpy as np
2
3 def construct_packing():
4     n = 26
5     centers = np.zeros((n, 2))
6
7     # Central circle
8     centers[0] = [0.5, 0.5]
9
10    # First ring (8 circles) at radius 0.2 from center
11    for i in range(8):
12        angle = 2 * np.pi * i / 8
13        centers[i + 1] = [0.5 + 0.2 * np.cos(angle), 0.5 + 0.2 * np.sin(angle)]
14
15    # Outer ring (13 circles) at radius 0.38 from center
16    for i in range(13):
17        angle = 2 * np.pi * i / 13
18        centers[9 + i] = [0.5 + 0.38 * np.cos(angle), 0.5 + 0.38 * np.sin(angle)]
19
20    # Add four corner circles
21    corners = [
22        [0.15, 0.15],  # Bottom-left corner
23        [0.85, 0.15],  # Bottom-right corner
24        [0.85, 0.85],  # Top-right corner
25        [0.15, 0.85]   # Top-left corner
26    ]
27    centers[22:26] = corners
28
29    radii = compute_max_radii(centers)
30    sum_radii = np.sum(radii)
31
32    return centers, radii, sum_radii
33
34 def compute_max_radii(centers):
35    n = centers.shape[0]
36    radii = np.zeros(n)
37
38    # Edge distances
39    edge_dists = np.minimum(centers[:, 0], centers[:, 1])
40    edge_dists = np.minimum(edge_dists, np.minimum(1 - centers[:, 0], 1 - centers[:, 1]))
41
42    # Nearest neighbor distances
43    min_neighbor_dists = np.full(n, np.inf)
44    for i in range(n):
45        for j in range(i + 1, n):
46            dist = np.linalg.norm(centers[i] - centers[j])
47            if dist < min_neighbor_dists[i]:
48                min_neighbor_dists[i] = dist
49            if dist < min_neighbor_dists[j]:
50                min_neighbor_dists[j] = dist
51
52    # Compute radii as the minimum between edge and half neighbor distance
53    radii = np.minimum(edge_dists, min_neighbor_dists / 2)
54
55    return radii
```

*Listing 8.* Circle packing implementation at iteration 55: exploratory ring-based pattern.

```python
import numpy as np

def construct_packing():
    n = 26
    centers = np.zeros((n, 2))

    # Central circle
    centers[0] = [0.5, 0.5]

    # First ring (8 circles) at radius 0.2 from center
    for i in range(8):
        angle = 2 * np.pi * i / 8
        centers[i + 1] = [
            0.5 + 0.2 * np.cos(angle),
            0.5 + 0.2 * np.sin(angle)
        ]

    # Outer ring (13 circles) at radius 0.35 from center (adjusted radius)
    for i in range(13):
        angle = 2 * np.pi * i / 13
        centers[9 + i] = [
            0.5 + 0.35 * np.cos(angle),
            0.5 + 0.35 * np.sin(angle)
        ]

    # Add four corner circles placed closer to the edges (adjusted positions)
    corners = [
        [0.1, 0.1],   # Bottom-left corner
        [0.9, 0.1],   # Bottom-right corner
        [0.9, 0.9],   # Top-right corner
        [0.1, 0.9]    # Top-left corner
    ]
    centers[22:26] = corners

    # Simulated annealing optimization
    best_centers = centers.copy()
    best_sum = compute_sum(best_centers)
    current_centers = best_centers.copy()
    current_sum = best_sum
    temp = 1.0
    cooling = 0.9995
    steps = 200000  # Increased steps for better optimization

    for _ in range(steps):
        idx = np.random.randint(n)
        delta = np.random.uniform(-0.015, 0.015, 2)  # Slightly larger delta for exploration
        new_center = current_centers[idx] + delta
        new_center = np.clip(new_center, 0.0, 1.0)
        new_centers = current_centers.copy()
        new_centers[idx] = new_center
        new_sum = compute_sum(new_centers)

        # Metropolis criterion with improved acceptance probability
        if new_sum > current_sum or np.random.rand() < np.exp((new_sum - current_sum) / temp):
            current_centers = new_centers.copy()
            current_sum = new_sum
            if new_sum > best_sum:
                best_centers = new_centers.copy()
                best_sum = new_sum
```

```
60          temp *= cooling
61
62      return best_centers, compute_max_radii(best_centers), best_sum
63
64  def compute_max_radii(centers):
65      n = centers.shape[0]
66      radii = np.zeros(n)
67
68      # Edge distances to all four sides
69      edge_dists = np.minimum(centers[:, 0], centers[:, 1])
70      edge_dists = np.minimum(edge_dists, np.minimum(1 - centers[:, 0], 1 - centers
        [:, 1]))
71
72      # Nearest neighbor distances using efficient computation
73      dist_matrix = np.linalg.norm(centers[:, np.newaxis] - centers, axis=2)
74      np.fill_diagonal(dist_matrix, np.inf)
75      min_neighbor_dists = np.min(dist_matrix, axis=1)
76
77      # Radii are the minimum between edge distance and half the nearest neighbor
        distance
78      radii = np.minimum(edge_dists, min_neighbor_dists / 2)
79
80      return radii
81
82  def compute_sum(centers):
83      return np.sum(compute_max_radii(centers))
```

*Listing 9.* Circle packing implementation at iteration 93: ring-based initialization with extensive simulated annealing.

## C. EditBench Case Study

We show the full code of the case study discussed in Section 3.6. Listing 10 is the given input. Listing 11 and Listing 12 demonstrate how SEMREP takes advantage of the semantics-preserving transformation and then performs the order-consistent parallelism by using `asyncio.gather`. Listing 13 and Listing 14 show how existing methods fail to preserve output ordering while successfully performing the parallelism optimization by using `asyncio.as_completed`.

```
1  import logging
2  import os
3  from typing import Any, Dict, List
4  from pydantic import BaseModel, Field
5  from carvana_enzo_worker.enums.gpt_enums import GptModels, VertextAIModels
6  from carvana_enzo_worker.providers.vertexai_claude_provider import
       VertexAIClaudeProvider
7  from carvana_enzo_worker.providers.vertexai_gemini_provider import
       VertexAIGeminiProvider
8  from carvana_enzo_worker.providers.azure_o1_provider import AzureOpenAIo1Provider
9  from carvana_enzo_worker.providers.azure_gpt_provider import
       AzureOpenAIChatProvider
10
11 # pylint: disable=W1203, C0415 [Use %s formatting in logging function, import-
       outside-toplevel]
12
13
14 class LLMArena(BaseModel):
15     """
16     A tool to generate chats using multiple LLM's for a given prompt
17     """
18
19     prompt: str = Field(..., description="The input prompt for the LLMs.")
20     models: List[str] = Field(..., description="A list of model names to use for
       generating chats.")
21     responses: List[str] = Field([], description="A list of generated chat
       responses.")
```

```
22     kwargs: Dict[str, Any] = Field({}, description="Additional keyword arguments
       for the LLMs.")
23
24
25     @staticmethod
26     async def generate_responses_for_models(prompt: str, models: List[str], **
       kwargs: Any) -> List[str]:
27         """
28         Generate responses from multiple models for a given prompt.
29
30         :param prompt: The input prompt for the LLMs.
31         :param models: A list of model names to use for generating responses.
32         :return: A list of generated responses.
33         """
34         responses = []
35         providers = []
36         for model in models:
37             provider_for_model = LLMArena._get_provider_for_model(model, **kwargs)
38             providers.append(provider_for_model)
39
40
41         for provider in providers:
42             try:
43                 response = await provider.generate_chat_response(prompt)
44                 responses.append(response)
45             except Exception as e:
46                 logging.error(f"Error generating response from {provider}: {e}")
47                 responses.append(f"Error generating response from {provider}: {e}")
48
49         return responses
50
51
52     @staticmethod
53     def _get_provider_for_model(model: str, **kwargs: Any) -> Any:
54         event_id = kwargs.get("event_id", "")
55
56         if model == VertextAIModels.CLAUDE_3_5_SONNET_V2.name:
57             return  VertexAIClaudeProvider(event_id=event_id, location=str(os.
       getenv("VERTEXAI_CLAUDE_REGION")), deployment_id=model)
58
59         if model == VertextAIModels.GEMINI_2_0_FLASH_EXP.name:
60             return VertexAIGeminiProvider(event_id=event_id, location=str(os.getenv
       ("VERTEXAI_GEMINI_REGION")), deployment_id=model)
61
62         if model == GptModels.o1.value:
63             return AzureOpenAIo1Provider(event_id=event_id, deployment_id=model)
64
65         return AzureOpenAIChatProvider(event_id=event_id, deployment_id=model)
```

*Listing 10.* EditBench input code.

```
1  --- sections/case_study/editbench/task_42_input.py   2026-02-24 14:58:47
2  +++ sections/case_study/editbench/task_42_ours_step1.py 2026-02-24 14:53:25
3  @@ -10,6 +10,7 @@
4
5   # pylint: disable=W1203, C0415 [Use %s formatting in logging function, import-
       outside-toplevel]
6
7  +import asyncio
8
9   class LLMArena(BaseModel):
10      """
11  @@ -37,15 +38,16 @@
12              provider_for_model = LLMArena._get_provider_for_model(model, **kwargs)
13              providers.append(provider_for_model)
```

```
14
15  -
16  -          for provider in providers:
17  +          async def generate_with_catch(provider, prompt):
18                 try:
19  -                     response = await provider.generate_chat_response(prompt)
20  -                     responses.append(response)
21  +                     return await provider.generate_chat_response(prompt)
22                 except Exception as e:
23  -                     logging.error(f"Error generating response from {provider}: {e}")
24  -                     responses.append(f"Error generating response from {provider}: {e
        }")
25  +                     msg = f"Error generating response from {provider}: {e}"
26  +                     logging.error(msg)
27  +                     return msg
28
29  +          responses = [await generate_with_catch(provider, prompt) for provider in
        providers]
30  +
31                 return responses
```

*Listing 11.* SEMREP extracts a helper function in Step 1 as a semantics-preserving refactor.

```
1  --- sections/case_study/editbench/task_42_ours_step1.py 2026-02-24 14:53:25
2  +++ sections/case_study/editbench/task_42_ours_step2.py 2026-02-24 14:52:59
3  @@ -46,8 +46,8 @@
4                     logging.error(msg)
5                     return msg
6
7  -          responses = [await generate_with_catch(provider, prompt) for provider in
        providers]
8  -
9  +          tasks = [generate_with_catch(provider) for provider in providers]
10 +          responses = await asyncio.gather(*tasks)
11             return responses
```

*Listing 12.* SEMREP parallelizes the helper calls using `asyncio.gather` in Step 2.

```
1  --- sections/case_study/editbench/task_42_input.py   2026-02-24 14:58:47
2  +++ sections/case_study/editbench/task_42_direct_step1.py 2026-02-24 14:57:36
3  @@ -37,15 +37,15 @@
4                 provider_for_model = LLMArena._get_provider_for_model(model, **kwargs)
5                 providers.append(provider_for_model)
6
7  -
8  -          for provider in providers:
9  +          tasks = [provider.generate_chat_response(prompt) for provider in providers
        ]
10 +          for task in asyncio.as_completed(tasks):
11                 try:
12 -                     response = await provider.generate_chat_response(prompt)
13 +                     response = await task
14                     responses.append(response)
15                 except Exception as e:
16 +                     provider = providers[tasks.index(task)]
17                     logging.error(f"Error generating response from {provider}: {e}")
18                     responses.append(f"Error generating response from {provider}: {e
        }")
19 -
20             return responses
```

*Listing 13.* The direct transformation parallelizes using `asyncio.as_completed` without preserving output order and misses the necessary import.

```
1 --- sections/case_study/editbench/task_42_direct_step1.py 2026-02-24 14:57:36
2 +++ sections/case_study/editbench/task_42_direct_step2.py 2026-02-24 14:59:23
3 @@ -9,8 +9,8 @@
4  from carvana_enzo_worker.providers.azure_gpt_provider import
     AzureOpenAIChatProvider
5
6  # pylint: disable=W1203, C0415 [Use %s formatting in logging function, import-
     outside-toplevel]
7 +import asyncio
8
9 -
10  class LLMArena(BaseModel):
11      """
12      A tool to generate chats using multiple LLM's for a given prompt
```

*Listing 14.* The direct transformation resolves the missing import.

# D. KernelBench Case Study

We compare SEMREP results with the top solutions from the KernelBench leaderboard[1]. To eliminate hardware variance, we run all solutions on a local L40S GPU and compute speedup relative to the baseline measured on the same machine.

### D.1. Matmul with Diagonal Matrices

The task is to optimize a model that performs matrix multiplication of a diagonal matrix with another matrix. SEMREP achieves a $13.2\times$ speedup, while the leaderboard baseline (Claude-3.5 Sonnet) achieves a $12.2\times$ speedup.

**Memory coalescing.** SEMREP uses a 1D linear index that maps threads directly to adjacent memory addresses. This allows the GPU to "coalesce" multiple memory requests into a single transaction, maximizing throughput. Claude's 2D block mapping ($16\times16$) introduces extra coordinate arithmetic and can lead to less efficient memory access if matrix dimensions are not perfectly aligned.

**Allocation overhead.** SEMREP uses `torch::empty_like`, which allocates memory without clearing it. Claude uses `torch::zeros`, which performs a redundant memory pass to fill the tensor with zeros before the kernel runs. Since the kernel overwrites every value anyway, this extra pass in Claude-generated output wastes memory bandwidth.

**Kernel simplicity.** The 1D grid in SEMREP simplifies the index calculation ($idx = i \cdot M + j$), reducing the number of integer instructions per thread compared to the 2D logic ($row$ and $col$ calculations) used in the Claude-generated output.

```python
1 import torch
2 import torch.nn as nn
3 from torch.utils.cpp_extension import load_inline
4
5 diag_matmul_cuda_source = """
6 #include <torch/extension.h>
7 #include <cuda_runtime.h>
8
9 __global__ void diag_matmul_kernel(
10     const float* diag,
11     const float* mat,
12     float* out,
13     const int N,
14     const int M) {
15
16     const int row = blockIdx.y * blockDim.y + threadIdx.y;
17     const int col = blockIdx.x * blockDim.x + threadIdx.x;
18
19     if (row < N && col < M) {
```

---

[1]https://scalingintelligence.stanford.edu/KernelBenchLeaderboard

```
20          out[row * M + col] = diag[row] * mat[row * M + col];
21      }
22  }
23
24  torch::Tensor diag_matmul_cuda(torch::Tensor diag, torch::Tensor mat) {
25      const int N = diag.size(0);
26      const int M = mat.size(1);
27
28      auto out = torch::zeros({N, M}, mat.options());
29
30      const dim3 threads(16, 16);
31      const dim3 blocks((M + threads.x - 1) / threads.x,
32                        (N + threads.y - 1) / threads.y);
33
34      diag_matmul_kernel<<<blocks, threads>>>(
35          diag.data_ptr<float>(),
36          mat.data_ptr<float>(),
37          out.data_ptr<float>(),
38          N, M);
39
40      return out;
41  }
42  """
43
44  diag_matmul_cpp_source = """
45  torch::Tensor diag_matmul_cuda(torch::Tensor diag, torch::Tensor mat);
46  """
47
48  diag_matmul = load_inline(
49      name='diag_matmul',
50      cpp_sources=diag_matmul_cpp_source,
51      cuda_sources=diag_matmul_cuda_source,
52      functions=['diag_matmul_cuda'],
53      verbose=True
54  )
55
56  class ModelNew(nn.Module):
57      def __init__(self):
58          super(ModelNew, self).__init__()
59          self.diag_matmul = diag_matmul
60
61      def forward(self, A, B):
62          return self.diag_matmul.diag_matmul_cuda(A.cuda(), B.cuda())
```

*Listing 15.* Claude-3.5 Sonnet for "Matmul with Diagonal Matrices".

```
1  import torch
2  import torch.nn as nn
3  from torch.utils.cpp_extension import load_inline
4
5
6  diag_mult_source = """
7   #include <torch/extension.h>
8   #include <cuda_runtime.h>
9
10
11   __global__ void diag_mult_kernel(const float* A, const float* B, float* out, int N
       , int M) {
12    int idx = blockIdx.x * blockDim.x + threadIdx.x;
13    if (idx < N * M) {
14    int i = idx / M;
15    int j = idx % M;
16    out[idx] = A[i] * B[idx]; // B is stored in row-major order (B[i*M + j] is B[i][j
       ])
17    }
```

```
18    }
19
20
21   torch::Tensor diag_mult_cuda(torch::Tensor A, torch::Tensor B) {
22     // Validate input dimensions
23     TORCH_CHECK(A.dim() == 1 && B.dim() == 2, "A must be 1D and B must be 2D");
24     TORCH_CHECK(A.size(0) == B.size(0), "A's length must match B's row count");
25
26
27     int N = A.size(0);
28     int M = B.size(1);
29
30
31     auto out = torch::empty_like(B);
32     const int threads_per_block = 256;
33     int blocks = (N * M + threads_per_block - 1) / threads_per_block;
34
35
36     diag_mult_kernel<<<blocks, threads_per_block>>>(
37     A.data_ptr<float>(),
38     B.data_ptr<float>(),
39     out.data_ptr<float>(),
40     N, M
41     );
42     return out;
43   }
44  """
45
46
47  diag_mult_cpp_source = (
48    "torch::Tensor diag_mult_cuda(torch::Tensor A, torch::Tensor B);"
49  )
50
51
52   # Compile the inline CUDA code
53  diag_mult = load_inline(
54    name="diag_mult",
55    cpp_sources=diag_mult_cpp_source,
56    cuda_sources=diag_mult_source,
57    functions=["diag_mult_cuda"],
58    verbose=True,
59  )
60
61
62  class ModelNew(nn.Module):
63    def __init__(self):
64      super().__init__()
65      self.diag_mult = diag_mult
66
67
68    def forward(self, A, B):
69      return self.diag_mult.diag_mult_cuda(A, B)
```

*Listing 16.* SEMREP for "Matmul with Diagonal Matrices".

## D.2. Conv2D Subtract Subtract Mish

The task is to optimize a model that performs a convolution, subtracts two scalar values element-wise, and applies Mish activation. SEMREP achieves $1.76\times$ speedup, while the leaderboard baseline (DeepSeek Coder) achieves $0.78\times$ slowdown.

**Kernel fusion.** DeepSeek Coder uses three separate CUDA kernels (elementwise subtract $\times2$, Mish), each requiring a kernel launch and writing intermediate results to global memory. SEMREP fuses the two subtracts and Mish into a single kernel. This reduces kernel launch overhead and avoids intermediate global-memory round-trips; values remain in registers

across the entire computation.

**Algebraic simplification.** DeepSeek Coder chains two subtract kernels: $x \rightarrow x - v_1 \rightarrow (x - v_1) - v_2$. SEMREP recognizes the semantic equivalence $(x - v_1) - v_2 = x - (v_1 + v_2)$ and applies a single offset, combining both subtractions into one operation inside the fused kernel.

**Allocation overhead.** Both elementwise-subtract and Mish kernels in DeepSeek Coder use `torch::zeros_like`, which zero-initializes memory before the kernel overwrites it. SEMREP uses `torch::empty_like`, which allocates without initialization, eliminating redundant memory writes since every element is overwritten by the kernel.

```python
 1  import torch
 2  import torch.nn as nn
 3  import torch.nn.functional as F
 4  from torch.utils.cpp_extension import load_inline
 5
 6  # Define the custom CUDA kernel for element-wise subtraction
 7  elementwise_subtract_source = """
 8  #include <torch/extension.h>
 9  #include <cuda_runtime.h>
10
11  __global__ void elementwise_subtract_kernel(const float* x, float* out, float value
        , int size) {
12      int idx = blockIdx.x * blockDim.x + threadIdx.x;
13      if (idx < size) {
14          out[idx] = x[idx] - value;
15      }
16  }
17
18  torch::Tensor elementwise_subtract_cuda(torch::Tensor x, float value) {
19      auto size = x.numel();
20      auto out = torch::zeros_like(x);
21
22      const int block_size = 256;
23      const int num_blocks = (size + block_size - 1) / block_size;
24
25      elementwise_subtract_kernel<<<num_blocks, block_size>>>(x.data_ptr<float>(),
        out.data_ptr<float>(), value, size);
26
27      return out;
28  }
29  """
30
31  elementwise_subtract_cpp_source = "torch::Tensor elementwise_subtract_cuda(torch::
        Tensor x, float value);"
32
33  # Compile the inline CUDA code for element-wise subtraction
34  elementwise_subtract = load_inline(
35      name='elementwise_subtract',
36      cpp_sources=elementwise_subtract_cpp_source,
37      cuda_sources=elementwise_subtract_source,
38      functions=['elementwise_subtract_cuda'],
39      verbose=True,
40      extra_cflags=[''],
41      extra_ldflags=['']
42  )
43
44  # Define the custom CUDA kernel for Mish activation
45  mish_source = """
46  #include <torch/extension.h>
47  #include <cuda_runtime.h>
48  #include <math.h>
49
50  __global__ void mish_kernel(const float* x, float* out, int size) {
```

```
51      int idx = blockIdx.x * blockDim.x + threadIdx.x;
52      if (idx < size) {
53          float exp_x = expf(x[idx]);
54          float softplus = logf(1.0f + exp_x);
55          out[idx] = x[idx] * tanhf(softplus);
56      }
57  }
58
59  torch::Tensor mish_cuda(torch::Tensor x) {
60      auto size = x.numel();
61      auto out = torch::zeros_like(x);
62
63      const int block_size = 256;
64      const int num_blocks = (size + block_size - 1) / block_size;
65
66      mish_kernel<<<num_blocks, block_size>>>(x.data_ptr<float>(), out.data_ptr<float
        >(), size);
67
68      return out;
69  }
70  """
71
72  mish_cpp_source = "torch::Tensor mish_cuda(torch::Tensor x);"
73
74  # Compile the inline CUDA code for Mish activation
75  mish = load_inline(
76      name='mish',
77      cpp_sources=mish_cpp_source,
78      cuda_sources=mish_source,
79      functions=['mish_cuda'],
80      verbose=True,
81      extra_cflags=[''],
82      extra_ldflags=['']
83  )
84
85  class ModelNew(nn.Module):
86      """
87      Optimized model that performs a convolution, subtracts two values, applies Mish
         activation using custom CUDA kernels.
88      """
89      def __init__(self, in_channels, out_channels, kernel_size, subtract_value_1,
        subtract_value_2):
90          super(ModelNew, self).__init__()
91          self.conv = nn.Conv2d(in_channels, out_channels, kernel_size)
92          self.subtract_value_1 = subtract_value_1
93          self.subtract_value_2 = subtract_value_2
94          self.elementwise_subtract = elementwise_subtract
95          self.mish = mish
96
97      def forward(self, x):
98          x = self.conv(x)
99          x = self.elementwise_subtract.elementwise_subtract_cuda(x, self.
        subtract_value_1)
100         x = self.elementwise_subtract.elementwise_subtract_cuda(x, self.
        subtract_value_2)
101         x = self.mish.mish_cuda(x)
102         return x
```

*Listing 17.* DeepSeek Coder for "Conv2D Subtract Subtract Mish".

```
1  import torch
2  import torch.nn as nn
3  from torch.utils.cpp_extension import load_inline
4
5  class ModelNew(nn.Module):
```

```python
6      def __init__(self, in_channels, out_channels, kernel_size, subtract_value_1,
       subtract_value_2):
7          super(ModelNew, self).__init__()
8          self.conv = nn.Conv2d(in_channels, out_channels, kernel_size)
9          self.offset = subtract_value_1 + subtract_value_2  # Combined subtraction
       offset
10
11         # Define and load the fused subtract + Mish CUDA kernel
12         fused_kernel_source = """
13         #include <torch/extension.h>
14         #include <cuda_runtime.h>
15         #include <math_constants.h>
16
17         __global__ void fused_subtract_mish_kernel(
18             const float* input, float offset, float* output, int size
19         ) {
20             int idx = blockIdx.x * blockDim.x + threadIdx.x;
21             if (idx < size) {
22                 float temp = input[idx] - offset;
23                 float exp_temp = expf(temp);
24                 float softplus_val = log1pf(exp_temp);
25                 float tanh_sp = tanhf(softplus_val);
26                 output[idx] = temp * tanh_sp;
27             }
28         }
29
30         torch::Tensor fused_subtract_mish_cuda(torch::Tensor input, float offset) {
31             auto size = input.numel();
32             auto output = torch::empty_like(input);
33
34             const int threads_per_block = 256;
35             const int blocks_per_grid = (size + threads_per_block - 1) /
       threads_per_block;
36
37             fused_subtract_mish_kernel<<<blocks_per_grid, threads_per_block>>>(
38                 input.data_ptr<float>(), offset, output.data_ptr<float>(), size
39             );
40             return output;
41         }
42         """
43
44         fused_kernel_cpp = (
45             "torch::Tensor fused_subtract_mish_cuda(torch::Tensor input, float
       offset);"
46         )
47
48         self.fused_kernel = load_inline(
49             name="fused_subtract_mish",
50             cpp_sources=fused_kernel_cpp,
51             cuda_sources=fused_kernel_source,
52             functions=["fused_subtract_mish_cuda"],
53             verbose=False
54         )
55
56     def forward(self, x):
57         conv_output = self.conv(x)
58         # Apply fused subtract and Mish in a single CUDA kernel
59         return self.fused_kernel.fused_subtract_mish_cuda(conv_output, self.offset)
```

*Listing 18.* SEMREP for "Conv2D Subtract Subtract Mish".

# E. Repo-Level Case Study

We use SWE-Perf (He et al., 2025) for our repo-level case study. Specifically, we present the task `astropy__astropy-13734`, which involves optimizing the fixed-width ASCII table parsing code in the Astropy library.

SEMREP achieves 0.20 statistical performance gain (He et al., 2025) during the semantics-preserving improvement and 0.32 after the performance-oriented transformation, while DeepSeek-V3-Chat achieves 0.15 after two iterations of performance-oriented transformation.

Both SEMREP and DeepSeek-V3-Chat address a common bottleneck by caching processed lines instead of calling `self.process_lines(lines)` twice. The key difference lies in column width computation: DeepSeek-V3-Chat makes repeated passes over the rows, while SEMREP uses a single pass with sequential row-wise access, resulting in better cache locality.

```diff
1  diff --git a/astropy/io/ascii/fixedwidth.py b/astropy/io/ascii/fixedwidth.py
2  index 9a2d322..d5e5233 100644
3  --- a/astropy/io/ascii/fixedwidth.py
4  +++ b/astropy/io/ascii/fixedwidth.py
5  @@ -34,11 +34,14 @@ class FixedWidthSplitter(core.BaseSplitter):
6         delimiter = '|'
7
8         def __call__(self, lines):
9  -          for line in lines:
10 -              vals = [line[x.start:x.end] for x in self.cols]
11 -              if self.process_val:
12 -                  yield [self.process_val(x) for x in vals]
13 -              else:
14 +          process_val = self.process_val
15 +          if process_val:
16 +              for line in lines:
17 +                  vals = [line[x.start:x.end] for x in self.cols]
18 +                  yield [process_val(x) for x in vals]
19 +          else:
20 +              for line in lines:
21 +                  vals = [line[x.start:x.end] for x in self.cols]
22                    yield vals
23
24         def join(self, vals, widths):
25 @@ -91,10 +94,12 @@ class FixedWidthHeader(basic.BasicHeader):
26                 List of table lines
27
28             """
29 +          # Process lines once and cache the result to avoid reprocessing.
30 +          processed_lines = list(self.process_lines(lines))
31
32             # See "else" clause below for explanation of start_line and position_line
33 -          start_line = core._get_line_index(self.start_line, self.process_lines(
      lines))
34 -          position_line = core._get_line_index(self.position_line, self.
      process_lines(lines))
35 +          start_line = core._get_line_index(self.start_line, processed_lines)
36 +          position_line = core._get_line_index(self.position_line, processed_lines)
37
38             # If start_line is none then there is no header line.  Column positions
      are
39             # determined from first data line and column names are either supplied by
      user
40 @@ -130,7 +135,7 @@ class FixedWidthHeader(basic.BasicHeader):
41                     # column names.  Note that get_fixedwidth_params returns Python
42                     # slice col_ends but expects inclusive col_ends on input (for
43                     # more intuitive user interface).
44 -                  line = self.get_line(lines, position_line)
45 +                  line = self.get_line(processed_lines, position_line)
```

```
46                    if len(set(line) - {self.splitter.delimiter, ' '}) != 1:
47                        raise InconsistentTableError(
48                            'Position line should only contain delimiters and '
49 @@ -149,7 +154,7 @@ class FixedWidthHeader(basic.BasicHeader):
50                    self.col_ends = [x - 1 if x is not None else None for x in
       col_ends]
51
52                # Get the header column names and column positions
53 -              line = self.get_line(lines, start_line)
54 +              line = self.get_line(processed_lines, start_line)
55                vals, starts, ends = self.get_fixedwidth_params(line)
56
57                self.names = vals
58 @@ -238,17 +243,24 @@ class FixedWidthData(basic.BasicData):
59       """ Splitter class for splitting data lines into columns """
60
61       def write(self, lines):
62 -          vals_list = []
63 +          # Collect all values and compute column widths in one pass.
64            col_str_iters = self.str_vals()
65 +          vals_list = []
66 +          # Initialize widths with zeros or with header name lengths if present.
67 +          if self.header.start_line is not None:
68 +              widths = [len(col.info.name) for col in self.cols]
69 +          else:
70 +              widths = [0] * len(self.cols)
71 +
72            for vals in zip(*col_str_iters):
73                vals_list.append(vals)
74 +              for i, val in enumerate(vals):
75 +                  if len(val) > widths[i]:
76 +                      widths[i] = len(val)
77
78 +          # Set column widths.
79            for i, col in enumerate(self.cols):
80 -              col.width = max(len(vals[i]) for vals in vals_list)
81 -              if self.header.start_line is not None:
82 -                  col.width = max(col.width, len(col.info.name))
83 -
84 -          widths = [col.width for col in self.cols]
85 +              col.width = widths[i]
86
87            if self.header.start_line is not None:
88                lines.append(self.splitter.join([col.info.name for col in self.cols],
```

*Listing 19.* DeepSeek-V3 on SWE-Perf.

```
1 diff --git a/astropy/io/ascii/fixedwidth.py b/astropy/io/ascii/fixedwidth.py
2 index 9a2d322..aa8c29f 100644
3 --- a/astropy/io/ascii/fixedwidth.py
4 +++ b/astropy/io/ascii/fixedwidth.py
5 @@ -51,7 +51,7 @@ class FixedWidthHeader(basic.BasicHeader):
6
7            """
8
9 -          # See "else" clause below for explanation of start_line and position_line
10 -          start_line = core._get_line_index(self.start_line, self.process_lines(
       lines))
11 -          position_line = core._get_line_index(self.position_line, self.
       process_lines(lines))
12 -
13 +          # Preprocess lines once to avoid repeated processing
14 +          processed_lines = list(self.process_lines(lines))
15 +          start_line = core._get_line_index(self.start_line, processed_lines)
16 +          position_line = core._get_line_index(self.position_line, processed_lines)
```

```
17          # If start_line is none then there is no header line.  Column positions
      are
18          # determined from first data line and column names are either supplied by
      user
19          # or auto-generated.
20 @@ -243,10 +243,22 @@ class FixedWidthData(basic.BasicData):
21          for vals in zip(*col_str_iters):
22              vals_list.append(vals)
23
24 +          # Compute column widths
25 +          lengths = []
26 +          for vals in vals_list:
27 +              lengths_row = [len(v) for v in vals]
28 +              lengths.append(lengths_row)
29 +
30 +          max_data = [max(col) for col in zip(*lengths)]
31 +
32 +          if self.header.start_line is not None:
33 +              name_lengths = [len(col.info.name) for col in self.cols]
34 +              max_lengths = [max(d, n) for d, n in zip(max_data, name_lengths)]
35 +          else:
36 +              max_lengths = max_data
37 +
38          for i, col in enumerate(self.cols):
39 -              col.width = max(len(vals[i]) for vals in vals_list)
40 -              if self.header.start_line is not None:
41 -                  col.width = max(col.width, len(col.info.name))
42 +              col.width = max_lengths[i]
43
44          widths = [col.width for col in self.cols]
```

*Listing 20.* SEMREP performs the semantics-preserving transformation on SWE-Perf.

```
1 diff --git a/astropy/io/ascii/fixedwidth.py b/astropy/io/ascii/fixedwidth.py
2 index 9a2d322..bb47725 100644
3 --- a/astropy/io/ascii/fixedwidth.py
4 +++ b/astropy/io/ascii/fixedwidth.py
5 @@ -92,10 +92,10 @@ class FixedWidthHeader(basic.BasicHeader):
6
7          """
8
9 -          # See "else" clause below for explanation of start_line and position_line
10 -          start_line = core._get_line_index(self.start_line, self.process_lines(
      lines))
11 -          position_line = core._get_line_index(self.position_line, self.
      process_lines(lines))
12 -
13 +          # Preprocess lines once to avoid repeated processing
14 +          processed_lines = list(self.process_lines(lines))
15 +          start_line = core._get_line_index(self.start_line, processed_lines)
16 +          position_line = core._get_line_index(self.position_line, processed_lines)
17          # If start_line is none then there is no header line.  Column positions
      are
18          # determined from first data line and column names are either supplied by
      user
19          # or auto-generated.
20 @@ -243,11 +243,20 @@ class FixedWidthData(basic.BasicData):
21          for vals in zip(*col_str_iters):
22              vals_list.append(vals)
23
24 +          # Compute column widths in a single pass for efficiency
25 +          num_cols = len(self.cols)
26 +          max_lengths = [0] * num_cols
27 +          for vals in vals_list:
28 +              for i, val in enumerate(vals):
```

```
29 +                   val_len = len(val)
30 +                   if val_len > max_lengths[i]:
31 +                       max_lengths[i] = val_len
32 +
33           for i, col in enumerate(self.cols):
34 -              col.width = max(len(vals[i]) for vals in vals_list)
35 +              width = max_lengths[i]
36               if self.header.start_line is not None:
37 -                  col.width = max(col.width, len(col.info.name))
38 -
39 +                  width = max(width, len(col.info.name))
40 +              col.width = width
41           widths = [col.width for col in self.cols]
42
43           if self.header.start_line is not None:
```

*Listing 21.* SEMREP performs optimization on SWE-Perf.

## F. Test Coverage

SEMREP's equivalence check relies on available tests. Lower coverage weakens the semantic constraint imposed by EMI. To study this trade-off, we ablate coverage by removing a subset of verification tests, reducing line coverage to $<30\%$. As shown in Table 6, relaxed coverage lowers Pass@1 (55.56 vs. 57.41) but raises Pass@16 (69.80 vs. 66.67). This suggests an interesting observation: stricter coverage enforces tighter behavioral consistency and improves single-shot correctness, while relaxed coverage permits more exploratory but potentially semantics-breaking edits that benefit sample-level discovery. Additionally, we adopt a held-out evaluation (one test for verification, all others held-out) to quantify how well verified outputs generalize: 2.3% of Stage 1 outputs and 7.4% of Stage 2 outputs that pass verification fail the held-out tests. Under single-test verification, overall performance decreases to Pass@1 = 51.85 ($-5.56\%$) and Pass@16 = 64.87 ($-1.80\%$).

*Table 6.* Effect of test coverage during mid-training on EditBench.

| Line Coverage | Pass@1 ↑ | Pass@16 ↑ |
|---:|:---:|:---:|
| $<30\%$ | 55.56 | **69.80** |
| $\geq99\%$ (Ours) | **57.41** | 66.67 |

## G. Inference Efficiency

The direct transformation baseline uses the same two-iteration setting as SEMREP's two-stage pipeline, making the comparison compute-matched by design. To further quantify inference efficiency, we compare wall-clock time based on Figure 3: our end-to-end framework with 4 rollouts ($\sim$147.7s/problem) achieves comparable performance to Kevin-32B with 16 rollouts ($\sim$440.4s/problem), reducing the required compute budget by more than 66%.

## H. Budget Split Sensitivity

We train SEMREP with alternative budget allocations and evaluate on EditBench. As shown in Table 7, all configurations with $\alpha > 0$ consistently outperform the *Baseline* ($\alpha = 0$), confirming the value of the proposed decomposition. Our default $\alpha = 0.50$ achieves the strongest balance between Pass@1 and Pass@16.

*Table 7.* Effect of budget split on EditBench. $\alpha = 0$ is the extensively finetuned baseline with the same total budget.

| $\alpha$ | Mid-training Budget ($\mathcal{B}$) | Finetuning Budget ($\mathcal{B}$) | Pass@1 ↑ | Pass@16 ↑ |
|---|:---:|:---:|:---:|:---:|
| 0 (Baseline) | 0 | $\mathcal{B}$ | 53.70 | 55.56 |
| 0.25 | $\mathcal{B}/4$ | $3\mathcal{B}/4$ | 59.26 | 64.29 |
| 0.50 | $\mathcal{B}/2$ | $\mathcal{B}/2$ | 57.41 | 66.67 |
| 0.75 | $3\mathcal{B}/4$ | $\mathcal{B}/4$ | 56.48 | 61.04 |

# I. Cross-Language Generalization

We evaluate SEMREP 's generalizability across programming languages by reporting per-language results on EditBench. As shown in Table 8, SEMREP outperforms the baseline on both Python and JavaScript.

*Table 8.* Per-language breakdown on EditBench.

| Model | Python ↑ | JavaScript ↑ |
|---|---|---|
| Baseline | 54.64 | 45.45 |
| **SEMREP (Ours)** | **57.73** | **54.55** |

# J. Readability Analysis

Following prior work (Mi et al., 2018; Li et al., 2025), we further investigate readability using character-, token-, and embedding-level similarities. As shown in Table 9, the semantics-preserving transformation of SEMREP led to more conservative transformations than the baseline in the first stage during inference. However, the final readability after the instruction-specific transformation does not differ much between SEMREP and the baseline, implying that SEMREP's approach does not degrade readability even without explicit instructions to preserve it.

*Table 9.* Readability analysis using character/token/embedding similarities (Mi et al., 2018; Li et al., 2025).

| | SEMREP (Ours) | | | Baseline | | |
|---|---|---|---|---|---|---|
| | $C_{src} \to C_{rep}$ | $C_{rep} \to C_{tgt}$ | $C_{src} \to C_{tgt}$ | $C_{src} \to C_{rep}$ | $C_{rep} \to C_{tgt}$ | $C_{src} \to C_{tgt}$ |
| Character | 0.306 | 0.099 | 0.118 | 0.132 | 0.251 | 0.108 |
| Token | 0.443 | 0.228 | 0.296 | 0.384 | 0.329 | 0.341 |
| Embedding | 0.987 | 0.986 | 0.982 | 0.983 | 0.991 | 0.982 |

