# OpenReview forum: "SemRep : Generative Code Representation Learning with Code Transformations"
_ICML.cc/2026/Conference — ICML 2026 regular_

### Official Review · Reviewer_kntd · 2026-03-06

**Soundness:** 3
**Presentation:** 4
**Significance:** 3
**Originality:** 3
**Overall Recommendation:** 5
**Confidence:** 4

**Summary:**

This paper proposes SEMREP, a framework designed to improve code transformation through generative code representation learning. The core idea of this approach is to employ semantics-preserving transformations as an intermediate representation. The authors propose that this representation can be used both to train the model as a generative task and to guide subsequent instruction-specific code transformations. The paper evaluates this framework on general code editing and CUDA kernel optimization tasks.

**Compliance With Llm Reviewing Policy:**

Affirmed.

**Final Justification:**

The authors have provided substantive new data and clarifications that have fully resolved my initial concerns, and my confidence in the work has been thoroughly reinforced. I firmly believe this paper is technically solid, advances the sub-area, and will be a valuable addition to the conference program.

**Key Questions For Authors:**

1. When the test coverage of the input code is extremely low (e.g., 0%), how are SEMREP's reward mechanism and final code transformation performance specifically affected?

2. Considering the parallel sampling of $k=16$ during inference, how much overhead is added to SEMREP's end-to-end wall-clock time compared to the direct transformation baseline model?

3. Regarding the AlgoTune case study, how does SEMREP perform if the unmodified evaluation formula from the original literature is strictly used?

**Limitations:**

yes

**Strengths And Weaknesses:**

Strengths:

1. The paper introduces SEMREP that explicitly uses LLMs to capture code semantics by generating semantics-preserving transformations. This approach effectively shifts away from standard end-to-end learning—which relies on implicit latent representations—by making semantic understanding an explicit and verifiable generative objective.

2. The proposed method demonstrates impressive and consistent performance gains across both general and domain-specific code editing tasks. The authors provide solid evidence that explicit training on semantics-equivalent code enhances the model's robustness against semantics-preserving code perturbations (e.g., changes to docstrings or syntax), achieving better consistency than standard finetuned baselines.

3. The paper successfully demonstrates that the exploratory nature of SEMREP integrates exceptionally well with evolutionary coding agents like OpenEvolve.

Weaknesses:

1. To achieve its performance, SEMREP employs an iterative evolutionary search during the inference phase. During inference, the model undergoes $T=2$ iterations, generating $k=16$ parallel rollouts at each iteration. This approach of generating a large number of candidate codes and executing test cases during inference brings extremely high latency and computational costs.

2. In the AlgoTune case study, the authors admit that the absolute runtime dependency in the original evaluator introduced variance, making it an unstable proxy. Consequently, the authors modified the evaluator to define the fitness score directly as the speedup ratio relative to the baseline implementation for correct solutions, while assigning a score of zero if incorrect.

---

> ### Author Rebuttal · Authors · 2026-03-31
>
> We sincerely thank the reviewer for their time and valuable feedback!
>
> **Q1. Test coverage and its impact.**
>
> | line coverage| Pass@1 | Pass@16 |
> |----------|---------------|---------------|
> | <30% | 55.56 | 69.8 |
> | ≥99% | 57.41 | 66.67 |
>
> When the test coverage is extremely low (e.g., 0%), the EMI-based rewards degenerate to spurious rewards. Basically, there are no tests available to measure the semantics of the original code. All edits receive the same reward, making the reward uninformative.
>
> As the reviewer suggested, we ablate test coverage during training to further investigate reduced test coverage during mid-training, as it weakens the semantic constraints imposed by EMI and may not necessarily limit learning. Specifically, we remove some tests used to measure the semantics equivalence of the generated code, such that the test coverage obtained by these smaller test sets is significantly reduced (<30%). As a result, the mid-training is relaxed, so the generated transformed code is less constrained when tested for semantic equivalence.
> As shown in the table, this leads to lower Pass@1 (55.56 vs. 57.41) but higher Pass@16 (69.80 vs. 66.67). This suggests an interesting observation: relaxed coverage permits more exploratory, but potentially semantics-breaking edits, while stricter coverage enforces tighter behavioral consistency, as reflected in Pass@1.
>
> **Q2. Inference overhead.**
>
> We would like to clarify that the direct transformation baseline reported in the paper uses the same setting (i.e., two iterations) as SemRep's end-to-end framework (i.e., two steps of direct transformation, matching SemRep's two-stage pipeline). The comparison is therefore compute-matched by design.
>
> Additionally, we conducted an experiment to quantify SemRep's computational efficiency relative to the direct transformation baseline. As shown in Figure 3, our end-to-end framework with 4 rollouts (wall-clock time: ~147.7s / problem) achieves comparable performance to Kevin-32B with 16 rollouts (wall-clock time: ~440.4s / problem), reducing the required compute budget by more than 66%.
>
> **Q3. AlgoTune with an unmodified evaluation formula.**
>
> We adopted relative speedup because it is more robust to environmental variance in absolute runtime and is the standard metric in related benchmarks (e.g., KernelBench).
>
> We compare the performance reward calculated using the original and the modified formulas across all methods. The average difference is 0.05, with a standard deviation of 0.044. This gives a standardized z of 1.12, corresponding to a p-value of 0.26. Therefore, the difference is not statistically significant.
>
> We will include both formulas in the next version of the paper.

---

> > ### Author Rebuttal · Reviewer_kntd · 2026-04-03
> >
> > Thank you for the comprehensive rebuttal and the efficiency analysis.

---

> > > ### Author Response · Authors · 2026-04-04
> > >
> > > We sincerely thank the reviewer again for the valuable feedback and for taking the time to review our response. We are grateful for your support for the paper's acceptance.

---

### Official Review · Reviewer_yFn4 · 2026-03-09

**Soundness:** 2
**Presentation:** 3
**Significance:** 2
**Originality:** 3
**Overall Recommendation:** 4
**Confidence:** 3

**Summary:**

This paper presents SEMREP, a framework that improves LLM-based code transformation by explicitly learning semantics-preserving code representations as a generative mid-training task. The key idea is to train the model in two stages: first, learning to generate semantically equivalent variants of input code (verified via test execution), and second, fine-tuning for instruction-specific transformations like optimization or bug fixing. At inference time, the model alternates between generating equivalent representations and applying task-specific edits within an evolutionary search loop. SEMREP is evaluated on EditBench (general code editing) and KernelBench (CUDA kernel optimization), where it outperforms strong baselines including  closed-weight models.

**Compliance With Llm Reviewing Policy:**

Affirmed.

**Key Questions For Authors:**

1. How sensitive is SEMREP's performance to the B/2 budget split between representation learning and task-specific finetuning? Results with alternative splits (e.g., B/4 vs 3B/4) would clarify whether the decomposition itself matters or whether the specific allocation is critical. If the split is not very sensitive, it would strengthen the paper; if it is highly sensitive, it would raise concerns about practical tuning.

2. What fraction of the generated semantically equivalent programs are actually equivalent on inputs beyond the test set used for verification? Even an approximate analysis using held-out tests would help quantify the reliability of the EMI-based reward and address the soundness concern about test coverage dependence.

3. In the ablation, +SEMREP TTS alone sometimes outperforms +SEMREP-trained alone (e.g., on EditBench Pass@1). This suggests that much of the benefit may come from the inference-time structure rather than the learned representations. Could the authors provide a more detailed breakdown of when and why the training component contributes beyond what the inference strategy provides?

**Limitations:**

yes

**Strengths And Weaknesses:**

Strengths

1. The core insight that explicitly training models to generate semantically equivalent code before task-specific transformation helps disentangle what to preserve from what to change — is well-motivated and intuitive. The paper provides a compelling illustrative example (Figure 1) showing how direct transformation fails on CUDA kernel optimization because it misses the underlying computation pattern, while SEMREP's two-step approach correctly identifies and exploits it. The EditBench case study (Figure 4) further demonstrates this advantage concretely, where SEMREP preserves output ordering semantics that the direct approach breaks.

2. The experimental design is commendably fair. The authors enforce a strictly equal training budget between SEMREP and the finetuning-only baseline (B/2 for representation learning + B/2 for task-specific training vs. B for finetuning alone), which means observed improvements cannot be attributed to extra compute. The ablation study (Tables 3 and 4) cleanly isolates the contributions of the training component and the test-time scaling component, showing both contribute individually and combine effectively.

3. The evaluation is comprehensive and practically grounded. The paper covers two distinct application domains (general editing and CUDA optimization), multiple metrics (correctness, speedup, robustness, generalization), and includes cross-device generalization experiments (L40S to H200) that are highly relevant for real-world deployment. The integration with OpenEvolve on MST and circle packing problems demonstrates that SEMREP's exploratory behavior composes well with external agent frameworks, which broadens its practical applicability.

Weaknesses

1. The reliance on test-based equivalence is a fundamental limitation that the paper acknowledges but does not sufficiently address experimentally. If the test set has low coverage, the model may learn to produce "equivalent" code that only matches behavior on observed inputs. The paper provides no analysis of how test coverage affects the quality of learned representations, nor does it measure how often the generated semantically equivalent code is actually inequivalent on unseen inputs. Given that this is the foundation of the entire approach, this gap is significant.

2. The training budget split of B/2 for each stage is presented as a fixed design choice without justification or sensitivity analysis. It is unclear whether this 50-50 split is optimal or whether different ratios would yield substantially different results. Since the paper's fairness argument rests on this fixed budget, understanding how performance varies with the allocation ratio is important for assessing whether the improvement comes from the two-stage decomposition itself or simply from a fortunate budget split.

3. The inference procedure requires two iterations with k=16 parallel rollouts each, effectively doubling the inference compute compared to a single-stage approach with the same k. The paper does not clearly account for this cost when claiming efficiency gains. While Section 3.5 shows scaling curves, a direct comparison of total inference FLOPs or wall-clock time between SEMREP and baselines at equivalent performance levels is missing, making the "25% less inference compute" claim in the abstract difficult to verify.

4. The EditBench evaluation uses a relatively small test set (the "core set"), and the improvements, while consistent, are modest in absolute terms (e.g., 53.70 → 57.41 Pass@1 in Table 1). With such a small evaluation set, these differences may not be statistically significant. The paper does not report confidence intervals or statistical significance tests for any of the results, which weakens the strength of the empirical claims.

---

> ### Author Rebuttal · Authors · 2026-03-31
>
> We are grateful for your effort and time in leaving a thorough and constructive review!
>
> **W1/Q2. Test-based equivalence and coverage.**
> | line coverage| Pass@1 | Pass@16 |
> |----------|---------------|---------------|
> | <30%     | 55.56 | 69.8         |
> | ≥99%     | 57.41 | 66.67         |
>
> As the reviewer suggested, we ablate test coverage during training to further investigate reduced test coverage during mid-training, as it weakens the semantic constraints imposed by EMI, which may or may not necessarily limit the learning. Specifically, we remove some tests used to measure the semantics equivalence of the generated code, such that the test coverage obtained by these smaller test sets is significantly reduced (<30%). As a result, the mid-training is relaxed, so the generated transformed code is less constrained when tested for semantic equivalence. As shown in the table, this leads to lower Pass@1 (55.56 vs. 57.41) but higher Pass@16 (69.80 vs. 66.67). This suggests an interesting observation: relaxed coverage permits more exploratory, but potentially semantics-breaking edits, while stricter coverage enforces tighter behavioral consistency, as reflected in Pass@1.
>
> Meanwhile, to directly quantify equivalence beyond the verification test set (Q2), we additionally evaluated all outputs with the held-out setting. Specifically, we randomly select one test as verification and treat all remaining tests as held-out. As a result, 2.3% of the generated outputs from the first stage (semantics-preserving transformation) and 7.4% of the generated outputs from the second stage (instruction-specific transformation) that passed verification fail the held-out tests. Under the single-test verification setting, the overall performance on the full test set decreases to Pass@1=51.85 (-5.56%) and Pass@16=64.87 (-1.8%).
>
> **W2/Q1. Budget (B) split sensitivity.**
>
> | α | Stage 1 Budget | Stage 2 Budget | Pass@1 | Pass@16 |
> |---|---------|---------|--------|---------|
> | 0 (Baseline) | 0 | B | 53.70 | 55.56 |
> | 0.25 | B/4 | 3B/4 | 59.26 | 64.29 |
> | 0.50 (Ours) | B/2 | B/2 | 57.41 | 66.67 |
> | 0.75 | 3B/4 | B/4 | 56.48 | 61.04 |
>
> Following the reviewer’s suggestion, we train our model with alternative budget allocations and evaluate on EditBench.
> All configurations that include the representation learning stage (α > 0) consistently outperform the extensively finetuned baseline with strictly the same training budget (α = 0), particularly on Pass@16, confirming the value of the proposed decomposition.
> While our default split (α = 0.50) is not universally optimal across all metrics, it achieves the strongest overall balance between Pass@1 and Pass@16.
> We will include the study of varying budgets in the next version of the paper.
>
> **W3. Inference efficiency.**
>
> The 25% less inference compute is computed using the AlgoTune case study. SemRep achieves comparable results with significantly fewer iterations than DeepSeek-V3-Reasoner (14 v.s. 19).
>
> We would like to clarify that the direct transformation baseline reported in the paper uses the same setting (i.e., two iterations) as SemRep's end-to-end framework (i.e., two steps of direct transformation, matching SemRep's two-stage pipeline). The comparison is therefore compute-matched by design.
>
> As the reviewer suggested, we conducted an experiment to quantify SemRep's computational efficiency relative to the direct transformation baseline. As shown in Figure 3, our end-to-end framework with 4 rollouts (wall-clock time: ~147.7s / problem) achieves comparable performance to Kevin-32B with 16 rollouts (wall-clock time: ~440.4s / problem), reducing the required compute budget by more than 66%.
>
> **W4. Statistical significance.**
>
> While the Pass@1 improvement is not statistically significant, scaling to Pass@16 reduces variance across multiple samples per input and yields a statistically significant improvement (p = 0.029 < 0.05). We will include more statistical analysis in the revised paper.
>
> **Q3. Training/Inference decomposition.**
>
> To better understand the respective contributions of inference paradigm (+SemRep TTS) and representation learning (+SemRep‑trained), we performed a task‑level overlap analysis on EditBench Pass@1.
>
> Across EditBench, +SemRep TTS and +SemRep‑trained individually solved the same 57 tasks. Beyond this shared subset, the two variants exhibit complementary strengths:
> +SemRep TTS uniquely solves 3 tasks that +SemRep‑trained fails to resolve;
> +SemRep‑trained uniquely solves 4 tasks that +SemRep TTS cannot solve.
>
>
> According to our preliminary study, solutions produced by +SemRep TTS are typically cosmetic or local refactoring (e.g., replacing `os.listdir` with `os.scandir`, removing redundant comments) without introducing more aggressive algorithmic changes.
> In contrast, tasks solved only by +SemRep‑trained models more often involve semantic or algorithmic transformations, such as replacing an N‑item loop with a small number of atomic OS‑level calls.

---

> > ### Author Rebuttal · Reviewer_yFn4 · 2026-04-08
> >
> > The authors have provided substantive new experiments that directly address each of my core concerns.

---

### Official Review · Reviewer_hwfj · 2026-03-13

**Soundness:** 3
**Presentation:** 4
**Significance:** 3
**Originality:** 3
**Overall Recommendation:** 5
**Confidence:** 4

**Summary:**

This paper proposes SEMREP, a code transformation framework that explicitly learns semantic representations by generating semantics-preserving code variants and then using those variants to guide downstream code editing and CUDA kernel optimization. The paper argues that prior code transformation methods either treat editing as a purely end-to-end task, with semantic reasoning hidden inside model weights, or depend on compiler-style intermediate representations that are expensive to build and may not align well with how code LLMs are pretrained. SEMREP decomposes the task into source code → semantic representation → transformed code, where the intermediate representation is a program that is semantically equivalent to the input under a semantic test set. In stage 1, the model is trained with GRPO-based reinforcement learning to generate diverse semantics-preserving variants, rewarded for compiling and matching the original behavior on semantic tests. In stage 2, the model is further trained for instruction-specific transformation, with rewards for compilation, semantic preservation, and satisfying the requested edit. At inference time, SEMREP alternates between generating equivalent variants and instruction-following transformations, samples multiple candidates, and selects the best ones through test-based verification, turning code editing into a verifiable search process rather than a one-shot edit. The paper evaluates SEMREP on EditBench for real-world instructed code edits and KernelBench for PyTorch-to-CUDA kernel optimization. SEMREP outperformed  other LLMs in terms of Correctness, Speedup and fast metrics.

**Compliance With Llm Reviewing Policy:**

Affirmed.

**Final Justification:**

The rebuttal phase addressed my concerns with strong analysis. So I raise the score for this paper.

**Key Questions For Authors:**

1. Can this pipeline be generalize to other programming languages?
2. How do authors ensure about the correctness of translated program in terms of readability for the LLM's generation provided from Stage 1 and Stage 2.
3. In figure 2, authors

**Limitations:**

Lack of correctness evaluation on aspects such as code readability.

**Strengths And Weaknesses:**

Strength:
- Paper is well-written with clear presentation, including good overview pictures.
- Experiments show impressive outperformance of SEMREP over LLMs in terms of correctness.
- Selecting well-known datasets for the experiment.
- The motivation of GRPO and defining the reward functions make sense.

Weakness:
- The idea of applying GRPO for code translation has been established [1]. Authors need to highlight what are the novelties of their proposed reward functions compared to the existing work.
- The selection of LLMs as baselines lacks several well-known LLMs, such as Gemini or Claude (Table 6- although Gemini's result showed in other datasets' evaluation in Table 1. Besides, GPT-4o-mini is much more compact than GPT-5.2. Authors should highlight the reasons for not choosing these LLMs as baselines and mention the potential of applying these baselines.
- In terms of semantic correctness, this approach relied mainly on the test suite. However, there are several aspects of correctness, such as readability and naming conventions [2].

References.
1. Improving LLM-Generated Code Quality with GRPO. Maxime Robeyns.
2. CodeUltraFeedback: An LLM-as-a-Judge Dataset for Aligning Large Language Models to Coding Preferences
Martin Weyssow, Aton Kamanda, Xin Zhou, Houari Sahraoui

---

> ### Author Rebuttal · Authors · 2026-03-31
>
> We really appreciate your time and effort in leaving constructive comments!
>
> **W1: Novelty compared to other GRPO for code work.**
>
> Thank you for raising the concern.
> We want to clarify that SemRep's contribution is to disentangle the learning of the semantics of the code to be transformed from the task-specific transformation. It also benefits downstream tasks that require high-quality exploration (Sections 3.2 and 3.7) by providing intermediate code representations, rather than a new reward function to improve code quality [1]. As SemRep is explicitly trained on semantics-equivalent transformations, it has the potential to produce diverse intermediate programs that are not yet optimized but may include rewrites helpful for enabling future optimizations. This behavior is particularly amenable to evolutionary search frameworks, e.g., OpenEvolve (Section 3.7), which also benefits from exploring non-greedy intermediate states. All of this design is orthogonal to GRPO and can be complemented with various RL algorithms.
> We will make the distinction clearer in the revised paper.
>
> **W2: Missing well-known LLM baselines.**
>
> We include additional well-known LLMs from the official KernelBench leaderboard:
> | Models | Best Speedup | Best Correctness |
> |-|-|-|
> | gpt-o1 | 1.29 | 0.28 |
> | claude-3.5-sonnet | 1.27 | 0.14 |
> | gpt-4o | 1.25 | 0.44 |
> | deepseek-coder | 1.20 | 0.20 |
> | llama-3.1-405b | 1.14 | 0.28 |
> | gemini-1.5-flash | 1.09 | 0.07 |
> | llama-3.1-70b | 1.06 | 0.12 |
> | Ours | 2.87 | 0.93 |
>
> Due to budget and time constraints during the rebuttal period, we adopt results directly from the official leaderboard rather than re-running these models with our prompts and SemRep-TTS.
> We acknowledge that these results use each model's default configuration and may not reflect the full potential under optimized prompting.
> Empirically, our method achieves a 2.2× speedup and 2.1× higher correctness than the best competing model, suggesting that the advantage is not attributable solely to prompting differences.
>
> **W3/Q2/Limitation: Correctness beyond test suites, e.g., readability analysis regarding Stage 1/2 outputs.**
>
> | Dimension   | SemRep (C_src→C_rep) | SemRep (C_rep→C_tgt) | SemRep (C_src→C_tgt) | Baseline (C_src→C_rep) | Baseline (C_rep→C_tgt) | Baseline (C_src→C_tgt) |
> |-|-|-|-|-|-|-|
> |Character|0.306|0.099|0.118|0.132|0.251|0.108|
> |Token|0.443|0.228|0.296|0.384|0.329|0.341|
> |Embedding|0.987|0.986|0.982|0.983|0.991|0.982|
>
> In this work, our primary objective is to disentangle the learning of the intended behavior of the original code from the task-specific transformation, which operates under verifiable rewards, e.g., based on executable tests.
> We thus prioritize downstream performance metrics (editing success, speedup, correctness) that can be validated through execution.
> While there may not be a consensus on a quantifiable metric for readability, we agree that incorporating intermediate rewards for readability, e.g., via LLM-as-a-judge, is a promising direction for future work and would complement the execution-based verification.
>
> As the reviewer suggested, we further investigate readability using token-, character-, and embedding-level similarity as proxy measures, following [2, 3].
> As shown in the table, the semantics-preserving transformation of SemRep (char: 0.306) led to more conservative transformations than the baseline (char: 0.132) in the first stage during inference.
> However, the final readability after the instruction-specific transformation does not differ much between SemRep and the baseline, implying that SemRep’s approach does not degrade readability even without explicit instructions to preserve it.
>
> **Q1: Generalization to other programming languages.**
>
> ||Python|JavaScript |
> |-|-|-|
> | Baseline| 54.64|45.45|
> | SemRep| 57.73|54.55|
>
> Our evaluation already covers cross-language settings. KernelBench tasks involve both Python and CUDA, requiring the model to reason across language boundaries to optimize performance. EditBench includes both JavaScript and Python tasks. Regarding the reviewer's question, we further investigate the per-language breakdown for EditBench. As shown in the table, SemRep achieves consistent improvements across both languages.
>
> **Q3:** It seems there is an incomplete question regarding Figure 2. We would be happy to follow up and address the reviewer’s concern.
>
>
>
> [1] Robeyns, Maxime, and Laurence Aitchison. "Improving llm-generated code quality with grpo." arXiv preprint arXiv:2506.02211 (2025).
>
> [2] Mi, Qing, Jacky Keung, Yan Xiao, Solomon Mensah, and Yujin Gao. "Improving code readability classification using convolutional neural networks." Information and Software Technology 104 (2018): 60-71.
>
> [3] Li, Weichen, Albert Jan, Baishakhi Ray, Junfeng Yang, Chengzhi Mao, and Kexin Pei. "EDITLORD: Learning Code Transformation Rules for Code Editing." arXiv preprint arXiv:2504.15284 (2025).

---

> > ### Author Rebuttal · Reviewer_hwfj · 2026-04-05
> >
> > The authors addressed all of my concerns. Please include these experiments in the camera-ready version of the paper.

---

> > > ### Author Response · Authors · 2026-04-06
> > >
> > > We sincerely thank the reviewer for carefully reviewing our response. We really appreciate your recognition of our effort to address the concerns raised, and we are grateful for your support for the paper's acceptance.

---

### Official Review · Reviewer_PQ7M · 2026-03-13

**Soundness:** 3
**Presentation:** 3
**Significance:** 3
**Originality:** 3
**Overall Recommendation:** 5
**Confidence:** 4

**Summary:**

This paper investigates the code transformation. It proposes to improve code transformation through generative code representation learning. The main idea is to employ the semantics-preserving transformations as the intermediate representation. And it will use this intermediate representation to train the model as a generative tasking and guide the instructive-specific code transformation. The experiments shows that the proposed framework outperforms the closed weight baselines.

**Compliance With Llm Reviewing Policy:**

Affirmed.

**Final Justification:**

The rebuttal addresses my concerns. This paper is acceptable.

**Key Questions For Authors:**

- How can we ensure semantic-preserving? It would be best to provide some theoretical analysis.

- For the semantics-preserving, currently, semantics-preserving is only a conclusion based on the performance comparison of Pass@1. If it is theoretical semantics-preserving, the performance of Pass@1 alone is still not enough.

**Limitations:**

yes

**Strengths And Weaknesses:**

+ The techniques presented in this paper are reasonable and feasible. It proposes to disentangle what must be preserved for what can be changed. Then it defines a mid-training task, generative code representation learning, by training the model to generate semantically equivalent programs. And then during the inference phase, given the input code, the trained model can generate the semantically equivalent code or transformed code following the editing instructions.

+ The presentation of this paper and the overview figures are pretty well to make the idea very clear and easy  to be understood.

+ The extensive experiments which are designed to follow the rationale of the methodologies show the performance of the proposed approach.

---

> ### Author Rebuttal · Authors · 2026-03-31
>
> We really appreciate your time and effort in reviewing our paper and giving us constructive comments!
>
> **Q1: Theoretical analysis for semantic preservation.**
>
> | line coverage | Pass@1 | Pass@16 |
> |----------|---------------|---------------|
> | <30%     | 55.56 | 69.8         |
> | ≥99%     | 57.41 | 66.67         |
>
> Great suggestion. To determine the general semantic equivalence of two arbitrary programs is fundamentally not decidable (according to Rice’s theorem). In our paper, we thus adopted equivalence modulo inputs (EMI) to formalize the *approximate* semantic equivalence. In this setting, test coverage largely dictates the strictness of equivalence checking; higher coverage provides stronger empirical assurance of behavioral equivalence. A quick investigation of our training samples confirmed that the tests used during mid-training achieved ≥99% line coverage.
>
> That said, your suggestion inspired us to further investigate reduced test coverage during mid-training, as it weakens the semantic constraints imposed by EMI, which may or may not necessarily limit the learning. Specifically, we remove some tests used to measure the semantic equivalence of the generated code, resulting in significantly reduced test coverage (<30%). As a result, the mid-training is relaxed, so the generated transformed code is less constrained when tested for semantic equivalence while enjoying more aggressive exploration, leading to more diverse generated programs.
> As shown in the table, this leads to lower Pass@1 (55.56 vs. 57.41) but higher Pass@16 (69.80 vs. 66.67). This suggests an interesting observation: relaxed coverage permits more exploratory, but potentially semantics-breaking edits, while stricter coverage enforces tighter behavioral consistency, as reflected in Pass@1. This serves as an exciting future work.
>
> **Q2: Pass@1 alone is insufficient for theoretical semantic preservation.**
>
> We agree that Pass@1 alone is insufficient and does not provide a theoretical guarantee.
> As mentioned above, the program equivalence is undecidable, so we do not claim a theoretical guarantee while resorting to equivalence up to given (high-coverage) tests.
>
> Our studies above show how strongly this semantic constraint is imposed during training: higher coverage improves single-shot correctness (Pass@1), while lower coverage weakens the constraint and improves sample-level discovery (Pass@16).
> We will include this discussion in the next version of the paper.

---

> > ### Author Rebuttal · Reviewer_PQ7M · 2026-04-03
> >
> > Thanks for the rebuttal. Please include the discussion as mentioned in the next version of this paper. I think this paper is fully acceptable.

---

> > > ### Author Response · Authors · 2026-04-04
> > >
> > > We sincerely thank the reviewer again for the constructive suggestions and support for the paper's acceptance. Following the review's suggestion, we have also updated the [draft](https://anonymous.4open.science/r/SemRep-FFB7/draft.pdf) by incorporating the discussion into the paper's discussion section and adding the corresponding experiments to the appendix.

---

### Decision · Program_Chairs · 2026-04-30

**Decision:**

Accept (regular)

**Comment:**

This paper proposes SemRep, a framework for code transformation that explicitly separates semantic preservation from task-specific editing by learning semantics-preserving transformations as an intermediate representation. The core idea is clear, technically meaningful, and well aligned with the problem setting.

The review consensus is strongly positive. The four reviewers gave overall recommendations of 5, 5, 4, and 5, with solid confidence. Reviewers consistently highlighted the conceptual clarity of disentangling what must be preserved from what can be changed, the strength of the empirical results across both general code editing and CUDA/kernel optimization, and the care taken to keep the comparison compute-matched. The method is also viewed as practically meaningful because it appears to improve robustness to semantics-preserving perturbations and to integrate naturally with search-based inference.

The main concerns focused on whether semantic preservation is established beyond the verification tests, how much of the gain comes from representation learning versus inference-time search, sensitivity to the training-budget split, inference cost, and the breadth of baselines. The authors responded substantively. In particular, they provided additional coverage ablations, held-out test analysis, budget-split sensitivity experiments, compute-efficiency measurements, significance analysis, and a stronger baseline context. Importantly, all four reviewers explicitly marked their concerns as fully resolved in the rebuttal acknowledgement, and several final justifications now state that the paper is acceptable or technically solid.

Some rebuttal additions are nontrivial and should be incorporated into the final version, especially the held-out equivalence analysis, the efficiency discussion, and the decomposition of training versus test-time search contributions. However, in this case, the new material strengthens an already strong submission rather than rescuing a weak one. The core method, motivation, and main empirical case were already present, and the discussion period mainly addressed reasonable soundness questions. Overall, this is a strong paper with clear relevance to the conference and a well-supported positive review record.